# A Self-Supervised PINN for Inertial Pose and Dynamics Estimations

## Abstract

Accurate real-time monitoring of not only movements, but also internal joint moments or muscle forces that cause movement in unrestricted environments is key for many clinical and sports applications. A minimally obstrusive way to monitor movements is with wearable sensors, such as inertial measurement units, using the fewest sensors possible. Current real-time methods rely on supervised learning, where a ground truth dataset needs to be measured with laboratory measurement systems, such as optical motion capture, which then needs to be processed with methods that are known to introduce errors. There is a discrepancy between laboratory and real-world movements, and for analysing new motions, new ground truth data would need to be recorded, which is impractical. Therefore, we introduce SSPINNpose, a self-supervised physics-informed neural network that estimates movement dynamics, including joint angles and joint moments, from inertial sensors without the need for ground truth data for training. We run the network output through a physics model of the human body to optimize physical plausibility and generate virtual measurement data. Using this virtual sensor data, the network is trained directly on the measured sensor data instead of a ground truth. Experiments show that SSPINNpose is able to accurately estimate joint angles and joint moments at $8.7°$ and $4.9\,\mathrm{BWBH\%}$, respectively, for walking and running at up to speeds of $4.9\,\mathrm{m\,s^{-1}}$ at a latency of $3.5\,\mathrm{ms}$. We further show the versatility of our method by estimating movement dynamics for a variety of sparse sensor configurations and inferring the positions where the sensors are placed on the body.

## 1 Introduction

Understanding the biomechanics of injury-causing events is important for injury prevention. However, injuries seldom occur in controlled environments (Wallbank et al., 2024; Heiderscheit et al., 2005). Therefore, in-the-wild capturing of human movement dynamics, e.g. kinematics, joint torques, and ground reaction forces (GRFs), is desirable. Currently, the gold standard for capturing kinematics is optical motion capture (OMC), which is limited to a lab environment. In OMC, a person is fitted with reflective markers that are tracked by multiple cameras. Joint torques are estimated from the kinematics and force data, which are measured using force plates embedded into the floor, which further limits the environment. Applying the markers by hand is error-prone and the resulting kinematics can vary between different assessors (McGinley et al., 2009). Furthermore, different processing techniques can also lead to different results (Werling et al., 2022).

An alternative to the limited setting of OMC is the use of inertial measurement units (IMUs). These small, lightweight sensors can be worn during sports activities. Recent studies have explored methods that, based on inertial sensing, estimate poses (Yi et al., 2021; Van Wouwe et al., 2024; Von Marcard et al., 2017; Huang et al., 2018; Jiang et al., 2022; Roetenberg et al., 2013), forces (Tan et al., 2024) or full dynamics (Karatsidis et al., 2019; Dorschky et al., 2019; 2020; Yi et al., 2022; Li et al., 2021; Winkler et al., 2022). The dynamics estimations are either based on deep learning (Yi et al., 2021; Winkler et al., 2022; Dorschky et al., 2020), trajectory optimization (Dorschky et al., 2019; Li et al., 2021) or static optimization (Karatsidis et al., 2019). Current deep-learning methods rely on supervised learning, which requires labeled data for training and, therefore, inherit the limitations of OMC. As a practical example, motions like high-speed running or sprinting require a large recording area, and are absent in widely used public IMU datasets like DIP-IMU and TotalCapture Huang et al.

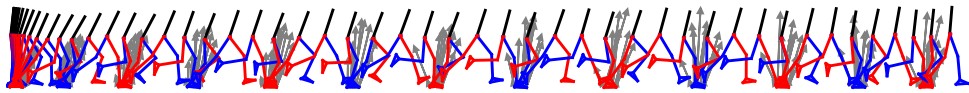

Figure 1: Example stickfigure of a running bout with a maximum speed of $4.9\,\mathrm{m\,s^{-1}}$ reconstructed with SSPINNpose. We show every the stick figure (black/red/blue) at intervals of $100\,\mathrm{ms}$ and the estimated GRFs (gray) every $20\,\mathrm{ms}$.

(2018); Trumble et al. (2017). Additionally, these datasets do not include force data. On the other hand, optimization-based methods need no labeled data but are computationally expensive. This makes them infeasible for analyzing dynamics over a long time period, which, for example, could be a running session leading to an injury. Both deep-learning and optimization-based methods can handle sparse IMU configurations (Winkler et al., 2022; Yi et al., 2021; Li et al., 2021; Dorschky et al., 2023), where not every body part is equipped with an IMU. This can make a system more practical for the user, but also makes the reconstruction of human movement dynamics even more challenging (Von Marcard et al., 2017). Similar to optical markers, the placement of IMUs can introduce errors in kinematic estimation. Therefore, inferring the sensor placement from the data can be highly beneficial.

To address these limitations, this work introduces SSPINNpose, which combines the real-time inference of learning-based methods with the ability of optimization-based approaches to reconstruct motions without relying on labeled data. The core principle behind SSPINNpose, a self-supervised physics-informed learning method, is that if an estimated motion is physically correct and corresponds to the measured IMU data, it is likely to be the correct motion. During training, the network is therefore guided to generate physically plausible motions that align with IMU data through virtual sensors. We further exploit auxiliary assumptions to accelerate training, mitigate local minima or enforce known properties of human movement.

Our main contribution is to transform the trajectory optimization problems from Li et al. (2021) and Dorschky et al. (2019) into self-supervised learning problem. We show that this method can also be used for real-time inference and with sparse IMU configurations. We further demonstrate that our method can be used to estimate the IMU placement. To our knowledge, SSPINNpose is the first real-time method for estimating biomechanical variables from inertial sensor data without labeled training data. An example of our model's output is shown in Figure 1.

## 2  RELATED WORK

Our work focuses on gait analysis, specifically the estimation of human movement dynamics, including both kinematics and the internal/external forces acting on the body. Since most dynamic motion during straight walking or running occurs in the lower limbs, particularly in the sagittal plane, we review works that either examine full-body motion or focus on this plane.

**Deep learning for movement dynamics:** In order to estimate the 3D pose of a person in real-time from sparse IMU configurations, Huang et al. (2018) proposed a deep learning-based method using a recurrent neural network (RNN). Subsequent work enhanced motion accuracy and allowed for flexible sensor configurations (Yi et al., 2021; Van Wouwe et al., 2024; Jiang et al., 2022; Zhang et al., 2024). Since visually plausible motion was prioritized in these early methods, physical correctness, such as accurate force estimation, became a significant next step. Therefore, Winkler et al. (2022) trained reinforcement learning agents to control torque-driven multibody dynamics models in a physical simulator. Another approach, Physical Inertial Poser (PIP), (Yi et al., 2022) introduced a physics module to create physically plausible motions. The physics module contains a proportional-derivative (PD) controller and a motion optimizer, which also yields joint torques and GRFs, but only the kinematics have been validated so far. Similar to PIP, two-stage inference methods, where kinematics are first estimated with a learned prior and then dynamically updated with a physics model, are established in the domain of video-based pose estimation Shimada et al. (2020); Xie et al. (2022); Kocabas et al. (2023); Yi et al. (2023); Yuan et al. (2021).

In biomechanical applications, reference data was often recorded with OMC in combination with force plates, from which joint angles and torques can be estimated via inverse kinematics and inverse kinematics. Supervised deep learning models have demonstrated to accurately predict these outcome variables from IMU data in a single inference step. Examples span a range of applications, such as gait analysis (Lim et al., 2019; Hernandez et al., 2021; Dorschky et al., 2020), slopes and stair climing (Chen et al., 2020), activities of daily living (Wang et al., 2023) or pediatric care Mohammadi Moghadam et al. (2024).

All deep-learning-based inertial pose and dynamics estimation methods to date rely on labeled data for training. Therefore, these methods are unable to predict out-of-distribution movements. Furthermore, supervised methods inherit limitations from the reference system that was used for labelling, which is usually OMC, such as eventual systemic biases or confinement to laboratory spaces. Our method requires no labeled data for training as we use a fully self-supervised approach.

**Optimization-based movement dynamics:** To estimate movement dynamics without labeled data, one can use optimization-based methods. Based on kinematics estimated by Xsens, Karatsidis et al. (2019) was first to propose the use of inverse methods to estimate GRFs and joint torques. From the estimated kinematics, they used static optimization to infer the GRF, and then used inverse dynamics to estimate the joint torques. They modeled the human body as a 3D musculoskeletal model with 39 degrees of freedom. However, their method has not been validated on running data and is not capable of real-time inference or handling sparse IMU setups. Furthermore, errors can accumulate during the multiple processing steps.

Movement dynamics can also be estimated in a single step with a trajectory optimization by finding control inputs, e.g. torques, for a simulation that best fits the IMU data. A solution to this problem can be found using optimal control. In optimal control, an objective function, in this case the distance between the actual and simulated IMU data, is minimized while satisfying dynamics constraints imposed by a multibody dynamics model. Dorschky et al. (2019) solved the resulting optimization problem with a two-dimensional musculoskeletal model with 9 degrees of freedom and 7 IMUs using a direct collocation method. However, they assumed the gait to be symmetric and periodic. Furthermore, they only optimised on averaged gait cycles data from multiple trials, while inference took more than 30 minutes for a single gait cycle. They later followed up with a study on sparse IMU configurations under the same settings (Dorschky et al., 2023). Optimal control problems with sparse IMU configurations under no symmetry assumptions have been solved by (Li et al., 2021), but they relied on the detection of gait events instead. Detecting gait events from IMU data is an additional error source and unreliable for fast motions. 3D optimal control problems based on IMU data of 3D movements have not been solved yet, except when synthetic IMU data was used (Nitschke et al., 2023).

Our method is conceptually related to optimal control, as we aim to find a motion that minimizes the distance between actual and simulated IMU data and is physically plausible. Unlike optimal control, we create a surrogate model to stochastically map inputs to outputs instead of solving discrete optimization problems as such. A further difference is that optimal control problems use physical correctness as a constraint, while we use it as an optimization objective instead. This is similar to the solving strategy of constraint relaxation in optimization. As our method relies on stochastic optimization through a deep learning model, we use first-order solvers, such as Adam (Kingma & Ba, 2017), instead of second-order solvers that are commonly used in optimal control problems, such as IPOPT (Wächter & Biegler, 2006). Our method is advantageous in terms of pre-processing, as we do not need to detect gait events (Li et al., 2021) or extract gait cycles under the assumption that these are periodic (Dorschky et al., 2019).

**From optimization problems to self-supervised learning:** Our work is based on the idea of transforming an optimization problem into a self-supervised learning problem. This have the advantage of speeding up the simulations, while not requiring labeled data. This approach has been used in various fields. For example, for 3D human (Schmidtke et al., 2023) and hand (Wan et al., 2019) shape matching, the shape of a hand or human body was predicted from a single image with a neural network. The shape, was then (neurally) rendered and compared to the input image. As the rendering process is differentiable, they can backpropagate the error to the neural network. A similar approach was used for the design of RF pulses in MRI (Jang et al., 2024), where the an optimal RF pulse prior was learned via MRI simulations. Self-supervised learning is also used in cloth simula-

tion, where the neural network predicts the mechanics of clothing during movement, which is then evaluated based on physical plausibility (Bertiche et al., 2021; Santesteban et al., 2022). Optimization problems are very specific to the task, which is also reflected in their respective self-supervised learning methods. In SSPINNpose, we reconstruct our input signal, comparable to Wan et al. (2019), and aim for physical plausibility as in Santesteban et al. (2022).

## 3 METHOD

### 3.1 PROBLEM FORMULATION

Our goal is to reconstruct lower body movement dynamics in the sagittal plane using IMUs. We aim to achieve this in a fully self-supervised manner, meaning that no labeled data for the outputs will be available during training.

The input consists of sequential two-dimensional accelerometer and gyroscope measurements from up to seven IMUs placed on the feet, shanks, thighs, and pelvis, alongside body constants that define the parameters of a multibody dynamics model. The outputs are the kinematics of the lower body, including root rotation and translations, joint angles, joint torques and GRF. All outcome parameters are directly estimated by the neural network, except for the GRFs, which are estimated with a ground contact model based on the kinematics output. We describe our method in the following section.

### 3.2 SSPINNPOSE

We introduce SSPINNpose, a self-supervised physics-informed neural network designed to learn human movement dynamics from IMU data without labels. The term "physics-informed" refers to the integration of Kane's equations and a temporal consistency loss, which ensures that the estimated velocities and accelerations align with changes in position and velocity over time. Temporal consistency describes that the velocities and accelerations are consistent with the changes in position and velocities, respectively. The self-supervised aspect relates to the reconstruction of the IMU data, allowing the model to learn from the inherent structure of the input signals. To ensure stable and fast training, we introduce further auxiliary losses that are based on either common assumptions in human movement or known properties of inertial sensors. In summary, SSPINNpose is trained with a weighted combination of the core (section 3.2.2) and auxiliary losses (section 3.2.3), which will be introduced in the following sections (see A for more details):

$$\mathcal{L} = \sum_{i \in \{IMU,T,K,GC\}} \lambda_i \mathcal{L}_i + \sum_{j \in \{B,\tau,slide,FS\}} \lambda_j \mathcal{L}_j \tag{1}$$

### 3.2.1 RNN IMPLEMENTATION

To capture the temporal dependencies inherent in human movements and inertial sensor data, we employ a recurrent neural network (RNN). We tested a LSTM (Hochreiter, 1997) for real-time inference and a bidirectional LSTM that has access to future information, each followed by two dense layers to calculate the output. At each time step $t$, the model receives the current IMU reading $\boldsymbol{x}_t$, body constants $\boldsymbol{\theta}_b$, IMU placement and rotations relative to their segment roots $\boldsymbol{\theta}_{imu}$, and ground contact model parameters $\boldsymbol{\theta}_{gc}$. The input IMU data consists of 2D acceleration and 1D angular velocity data per sensor in the sagittal plane, and is augmented with Gaussian noise with a standard deviation of $\eta_{imu}\sigma(\boldsymbol{x}_i)$ for each input channel $i$, where $\eta_{imu}$ is set to $0.25$

The 46 output features $\hat{\boldsymbol{y}}_t$ consist of the estimated generalized coordinates $\boldsymbol{q}$, velocities $\dot{\boldsymbol{q}}$, accelerations $\ddot{\boldsymbol{q}}$, torques $\boldsymbol{\tau}$ and ground contact model states, which consists of the global kinematics of the ankle joint $\tilde{\boldsymbol{q}}_{ankle}, \dot{\tilde{\boldsymbol{q}}}_{ankle}$, and a current friction factor for each foot $\hat{\boldsymbol{\mu}}$. We do not predict the horizontal position. For the loss calculations introduced in the following sections, we compute the global kinematics for the joints $\boldsymbol{p}_j$, IMUs $\boldsymbol{p}_{IMU}$ and ground contact points $\boldsymbol{p}_{gc}$ based on the kinematics of the respective parent joint. The global kinematics of each point consist of its global position, $x, y$, and angle, $\alpha$, as well as their first and second derivatives $\boldsymbol{p} = \{x, \dot{x}, \ddot{x}, y, \dot{y}, \ddot{y}, \alpha, \dot{\alpha}, \ddot{\alpha}\}$ (see A for further details).

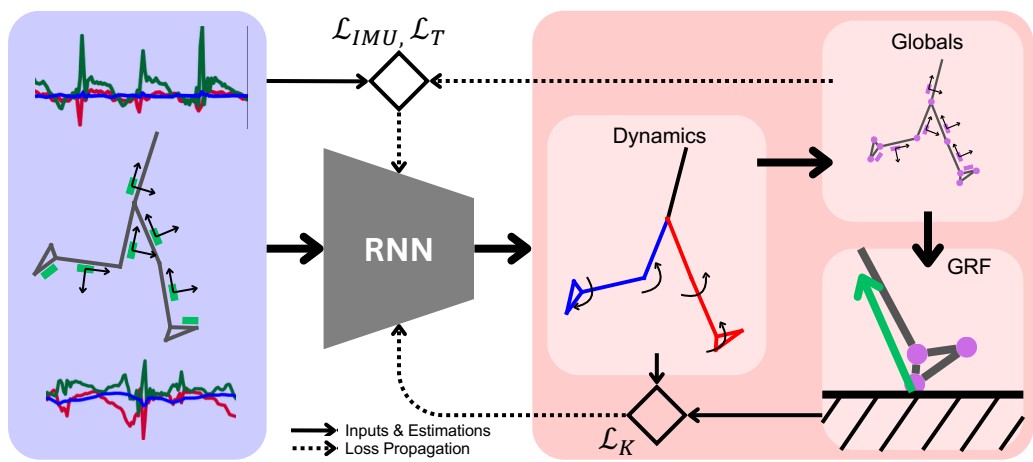

Figure 2: Overview of the SSPINNpose's training scheme. The blue box shows inertial measure unit (IMU) signals from an unknown motion. For simplicity, we only show a single pose (gray). IMUs are annotated in light green. The RNN estimates the multibody dynamics in the first light red box. We then calculate the global kinematics for all joints, virtual IMUs, the heels and the toes (magenta). The ground reaction force (GRF, green) is then estimated based on the global ankle kinematics. Then we calculate the IMU loss ($\mathcal{L}_{IMU}$) and the temporal consistency loss ($\mathcal{L}_T$) based on the global positions and Kane's Loss ($\mathcal{L}_K$) based on the estimated joint angles, torques and GRFs.

### 3.2.2 PHYSICS INFORMATION AND SELF-SUPERVISION

The main idea behind SSPINNpose is that a motion that is physically plausible and consistent with the IMU data is likely to be the correct motion. We enforce this by the following loss functions: Kane's loss ($\mathcal{L}_K$), temporal consistency loss ($\mathcal{L}_T$) and IMU reconstruction loss ($\mathcal{L}_{IMU}$). These core components of SSPINNpose are illustrated in Figure 2.

**Multibody Dynamics Model & Kane's Equations:**  Our multibody dynamics model is a sagittal-plane lower limb model with 2 translational and 7 rotational degrees of freedom, which correspond to the generalized coordinates. The body consists of 7 segments: one trunk, and a thigh, shank, and foot for each leg. The body constants contain the mass, length, center of mass and moment of inertia for each segment. The body constants are linearly scaled based on the participant's height (Winter, 2009). The forces scale linearly with the bodyweight, therefore, we set it to $1\,\mathrm{kg}$.

Using this dynamics model, we calculate the equations of motion based on Kane's method (Kane & Levinson, 1985), implemented in SymPy (Meurer et al., 2017). Kane's formulation is advantageous for deep learning as it is the method that requires fewer equations to be solved to describe movement dynamics. Kane defined that the sum of internal ($F_r^*$) and external ($F_r$) forces acting on a system is zero. Therefore, we can define a loss term that enforces the physical plausibility of each estimated state:

$$\mathcal{L}_K = |\boldsymbol{F}_r^* + \boldsymbol{F}_r| = f\left(\hat{\boldsymbol{y}}, \boldsymbol{\theta}_b, \boldsymbol{F}_{gc}\right). \qquad (2)$$

To estimate the GRF $\boldsymbol{F}_{gc}$, we model the foot-ground contact with a sliding contact point. The contact point's position between the heel and toe is determined based on the global ankle rotation. The vertical component of the GRF is modeled as a linear spring-damper system as in van den Bogert et al. (2011), while the horizontal component is modeled as a friction cone with a learned current friction coefficient $\hat{\mu}$. To disentangle the GRF from the kinematics, we estimate the global ankle kinematics seperately, which is supervised by the distance to the estimated forward kinematics ($\mathcal{L}_{GC}$) of the ankle. For more details, see A.

**Temporal Consistency Loss:**  While Kane's method enforces physical plausibility at each time point, we also ensure that the derivatives of the estimated coordinates match the estimated velocities, and that the derivatives of the estimated velocities match the estimated accelerations. This loss is

applied to the generalized coordinates $\boldsymbol{q}$. We normalize by the standard deviation of the estimated coordinates or velocities over the sequence to ensure that the loss is scale-invariant:

$$\mathcal{L}_T = \frac{1}{2n_{\boldsymbol{q}}} \sum_{i=1}^{n_{\boldsymbol{q}}} \left( \left( \frac{\delta \boldsymbol{q}_i}{\delta t} - \dot{\boldsymbol{q}}_i \right) \sigma(\boldsymbol{q}_i)^{-1} + \left( \frac{\delta \dot{\boldsymbol{q}}_i}{\delta t} - \ddot{\boldsymbol{q}}_i \right) \sigma(\dot{\boldsymbol{q}}_i)^{-1} \right). \tag{3}$$

We chose this approximate integration method to decouple the learning of kinematics from movement dynamics, as numerical differentiation of the kinematics would cause exploding gradients in Kane's equations.

**IMU Reconstruction Loss:** We obtain virtual IMU signals $\hat{\boldsymbol{x}}_{imu}$ by rotating the kinematics of each IMU $\boldsymbol{p}_{IMU}$ into its respective local coordinate system. These virtual IMU signals are then compared to the recorded IMU signals. We normalize by the standard deviation over a sequence of the IMU signals per channel and the number of IMUs $n_{imu}$:

$$\mathcal{L}_{IMU} = \frac{1}{n_{imu}} \sum_{i=1}^{n_{imu}} \left( \boldsymbol{x}_{imu} - \hat{\boldsymbol{x}}_{imu} \right) \sigma(\boldsymbol{x}_{imu})^{-1}. \tag{4}$$

### 3.2.3 Auxiliary Losses

This section describes the auxiliary losses that we use to accelerate training, mitigate local minima or enforce known properties of human movement. For more details and an abliation study to justify these losses, refer to the supplementary sections A and C.

**Joint Limit and Ground Contact Force Bounds** ($\mathcal{L}_B$): We penalize the model for exceeding joint limits and for violating bounds on maximum velocity and vertical position (see A). Additionally, we assume that for each sequence, each foot supports at least 20% of the body weight. In practice, this avoids local minima where the model does not predict any ground contact or skips on one foot.

**Torque Minimization** ($\mathcal{L}_\tau$): We apply a small weight on speed-weighted torque minimization, as minimizing effort is a common assumption in human movement and usually leads to more natural motions (van den Bogert et al., 2011). Similar to Dorschky et al. (2019), we normalize the torques by the maximum speed of the root translation in the sagittal plane. As our training data might contain some non-movement phases, the speed normalization only applies to sequences with estimated moving speeds greater than $1\,\mathrm{m\,s}^{-1}$.

**Sliding Penalty** ($\mathcal{L}_{slide}$): To prevent foot sliding when a ground reaction force (GRF) is present, we define sliding as the product of foot-ground speed and vertical GRF. This formulation ensures that at least one of these variables is constrained to be zero.

**Foot Speed** ($\mathcal{L}_{FS}$): To speed up the training process and make our model less susceptible to local minima, we make use of known properties of foot-worn IMUs by reconstructing their global velocities ($\dot{\boldsymbol{p}}_{K,x}$) using a Kalman filter with zero-velocity updates (Solà, 2017; Simon Colomar et al., 2012), as implemented in Küderle et al. (2024). This algorithm is based on integration of the IMU signals which accumulates errors from drift and noise. Furthermore, zero-velocity updates are unreliable during running. In consequence, we treat these reconstructed speeds as erroneous and only apply a penalty when the estimated foot-worn IMU speed from our kinematics differs by more than 30% from its reconstructed maximum speed during the sequence.

## 4 Experiments

In this section, we first describe the dataset used for training and evaluation, followed by the evaluation metrics used to assess our model's performance. Next, we show and discuss model's capability to estimate human movement dynamics from IMU data in section 4.1. Next, we show and discuss experiments regarding finetuning for physics and sensor placement personalizations (section 4.2) and sparse IMU configurations (section 4.3).

**Dataset** We use the "Lower-body Inertial Sensor and Optical Motion Capture Recordings of Walking and Running" dataset for training and evaluation (Dorschky et al., 2024). The dataset contains data of persons walking and running through an area equipped with OMC cameras and a single force plate, along with continuous IMU signals. For every trial, the OMC data contains roughly $5\,\mathrm{m}$ of kinematics data and force plate data for a single step. We downsampled the IMU signals to $100\,\mathrm{Hz}$. The dataset includes data from 10 participants, each performing 10 trials at 6 different speeds, ranging from $0.9\,\mathrm{m\,s^{-1}}$ to $4.9\,\mathrm{m\,s^{-1}}$. For each condition, the first 7 trials were designated for training, while the remaining 3 were used for evaluation.

We selected the training data by applying a heuristic that identifies standing and turning phases based on the foot and pelvis IMU signals, respectively. This was done to include the run-up to the motion capture area and some steps after the motion capture area in our training set, while avoiding turning phases that we cannot reconstruct with a two-dimensional model. In total, our training data consists of 76 minutes of unlabeled IMU data. We processed the OMC and force plate data with addBiomechanics (Werling et al., 2022) to compare the resulting joint angles and joint torques. The first participant was excluded from addBiomechanics because of erroneous force plate readings. During training, we randomly selected sequences of $256$ time steps from the training data, while full sequences were used during evaluation. Typical sequences from the datasets are visualized in Figures 1 and 7. This dataset has been used by several other works focussing on sagittal-plane lower limb dynamics (Dorschky et al., 2019; 2020; 2023).

**Metrics:** We use the following metrics to evaluate our model: *1.) Joint Angle Error (JAE):* The root mean square deviation (RMSD) between the estimated joint angles and those obtained from addBiomechanics, including the root orientation, in degrees. *2.) Joint Torque Error (JTE):* The RMSD between the estimated joint torques and those obtained from addBiomechanics, in bodyweight-bodyheight percent ($\mathrm{BWBH\%}$). *3.) GRF Error (GRFE):* The root mean square error (RMSE) between the estimated GRFs and those obtained from the force plate, normalized by the bodyweight, in bodyweight percent ($\mathrm{BW\%}$). The GRF is the only outcome variable that can be directly measured, therefore, we consider it to be an error and not a deviation to a reference system. *4.) Speed Error:* The RMSD between the estimated average speed and the sagittal-plane speed of the pelvis markers while the participant was crossing the OMC area, in $\mathrm{m\,s^{-1}}$. For all metrics, lower values are better. We show an evaluation on metrics that are commonly used in computer graphics in the supplementary B.

### 4.1 QUANTITATIVE AND QUALITATIVE EVALUATION

In the following, we show the performance of SSPINNpose on the test data (Table 1). We evaluated SSPINNposes performance on continuous IMU data using a LSTM and a Bi-LSTM model, respectively. Between both, there are only minor differences in the outcome metrics. The LSTM model estimated dynamics and GRFs slightly more accurately, while the Bi-LSTM model estimated speed more accurate and produces smoother motions. The LSTM can estimate the joint angles and torques in real-time, with a latency of $3.5\,\mathrm{ms}$. Training took approximately 16 hours on a NVIDIA RTX 3080 GPU. To compare against state of the art methods that report results on the same dataset, we show versions of our model with adapted training and evaluation schemes. To compare to Dorschky et al. (2019), which optimized on ensemble averaged gait cycles, we trained and evaluated SSPINNpose (OCP) on ensemble averaged gait cycles from all $60$ trials. For a fairer comparison to the regression-based Dorschky et al. (2020), we trained SSPINNpose (Reg) on continuous IMU data from 7 participants and evaluated on ensemble averaged gait cycles from the remaining 3 participants.

SSPINNpose's kinematics estimations are on par with current real-time deep learning-based methods (Yi et al., 2022) (see B for more details). Compared to existing biomechanically validated methods, SSPINNpose able to estimate the dynamics of human movement from IMU data in real-time without the need for labeled data. We achieve a speed error that is $0.1\,\mathrm{m\,s^{-1}}$ smaller the current optimal control-based state-of-the-art (Dorschky et al., 2023) when trained on continuous IMU data. The JAE, JTE, and GRFE, on the other hand, are generally larger than the CNN-based estimation from Dorschky et al. (2020), when tested under the same conditions. However, both the CNN and the optimal control-based methods took assumptions that are not applicable to real-time inference by segmenting the data into gait cycles. Furthermore, they assumed them to be symmetric and periodic,

Table 1: Quantitative comparison on continuous IMU data and on the test sets of Dorschky et al. (2019) and Dorschky et al. (2020). The best results are shown in bold. To compare against citet-dorschkyEstimationGaitKinematics2019 and Dorschky et al. (2020), we retrained and evaluated SSPINNpose in settings that are comparable to their methods.

| Model | JAE [deg] | JTE [BWBH%] | GRFE [BW%] | Speed [m s$^{-1}$] |
|---|---|---|---|---|
| SSPINNpose (LSTM) | 8.7 | 4.9 | 16.4 | 0.19 |
| SSPINNpose (Bi-LSTM) | 8.9 | 5.0 | 18.8 | 0.15 |
| SSPINNpose (OCP) | 8.9 | 6.8 | 23.8 | 0.25 |
| Dorschky et al. (2019) | 6.3 | 2.6 | 17.9 | 0.25 |
| SSPINNpose (Reg) | 11.2 | 7.2 | 23.1 | 0.12 |
| Dorschky et al. (2020) | 4.9 | 1.4 | 10.7 | - |

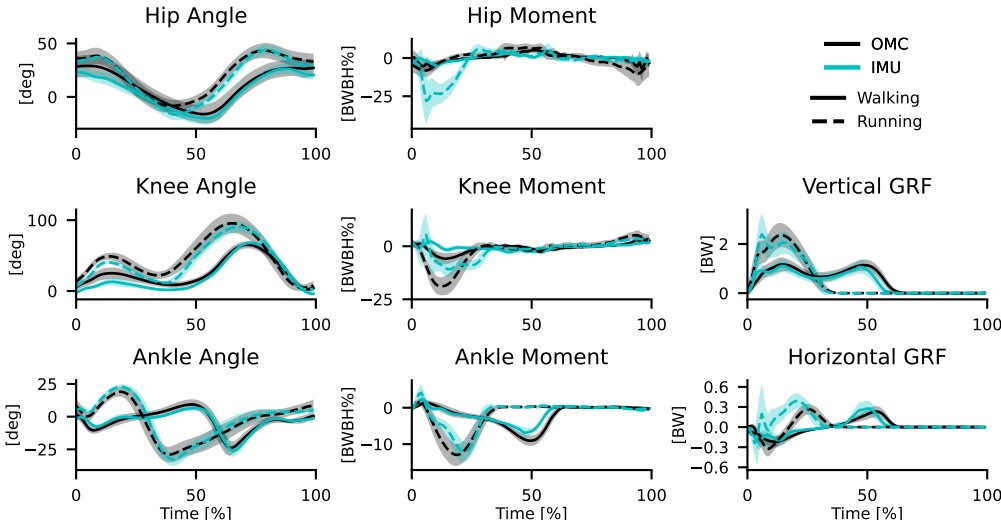

Figure 3: Average joint angles, torques and ground reaction forces (GRFs) for the right leg over all test gait cycles. Estimated with the Bi-LSTM. We segmented the gait cycles during which the force plate was hit and normalized them to a duration of 100 samples. Walking and running data is shown in solid and dashed lines, respectively. Our estimates are shown in cyan, the reference data is shown in black. The shaded area represents the standard deviation.

which limits the generalization towards arbitrary movements. SSPINNpose (Reg) was evaluated under a domain shift from noisy, continuous IMU data to averaged gait cycles. In our experiments, we found that torques and GRFs are more sensitive to the domain shift than joint angles. We argue that the comparison on continuous IMU data is a more realistic scenario and therefore more relevant, while the need for gait cycle segmentation is a limitation of the other methods.

In Figure 3, we show the gait-cycle averages of the joint angles, torques and GRFs estimated with the Bi-LSTM model in comparison to the OMC reference. The kinematics were estimated accurately, with a small bias in the hip and knee angle. Especially in running, the hip and knee moment were not accurately estimated during the stance phase, which is the first $40\,\%$ of the gait cycle for running and the first $60\,\%$ for walking. The ankle moment and vertical GRF shows slightly lower values than the reference data, while the horizontal GRF could not be estimated correctly. SSPINNpose estimated the kinematics and speeds robustly, with median and 95th percentile errors of $5.2°$ and $16.7°$ for joint angles, and $3.1\,\%$ and $9.3\,\%$ for speed.

In SSPINNpose, the network implicitly learns the interconnection between movement kinematics and dynamics. In contrast, other real-time capable methods (Yi et al., 2022; Shimada et al., 2020), use kinematical network output as an input signal for a PD controller and dynamics optimizer. Whether direct or two-stage inference is preferable, depends on the application. Direct inference

is faster and the dynamics are not subject to propagation errors from the kinematics. On the other hand, the two-stage inference can help with generalization towards unseen movements and can be more robust to noise. When testing SSPINNpose with PIP's PD controller and optimizer, we found that the the PD controller hyperparameters needed to be adjusted. We found that, depending on the hyperparameters, the PD controller could be used to slightly reduce the JTE while raising the JAE and smoothing the motion (see C for implementation details).

Our method contains a number of assumptions and simplifications. As in Yi et al. (2022), we assume that the ground is flat and the foot cannot slide. Information about the ground is present in the IMU data, and has been exposed in recent work (Jiang et al., 2022). The interaction between foot and ground is modeled as a linear spring-damper system. Furthermore, the multibody dynamics model is based on a generic template, which is due to a lack of personalization options. As we fit towards IMU signals that are noisy, our model can learn to replicate that noise and becomes less physically plausible, which we are mitigating, but not eliminating by augmenting the input data with Gaussian noise. Our model is able to accurately estimate human movement dynamics despite these limitations, therefore we consider them to be an opportunity to make the estimations more accurate in the future.

### 4.2 FINETUNING FOR PHYSICS AND PERSONALIZATION

In an ideal simulation, the estimated dynamics should perfectly match the actual motion. However, achieving a perfect simulation requires physical exactness, meaning that both Kane's loss and the temporal consistency loss must be zero. Therefore, we finetuned the Bi-LSTM towards physics by increasing the weight of the Kane's loss and the temporal consistency loss by a factor of 10. This reduced the JTE by 10% and the GRFE by 20%. However, as the IMU signals were not followed as strictly, the JAE increased by 5% and the speed error increased by 33%. After finetuning, the biases in knee moment and vertical GRF were substantially reduced and only the bias in the hip torque during the stance phase in running remained. For use cases where the torques are of most interest, this trade-off should be acceptable.

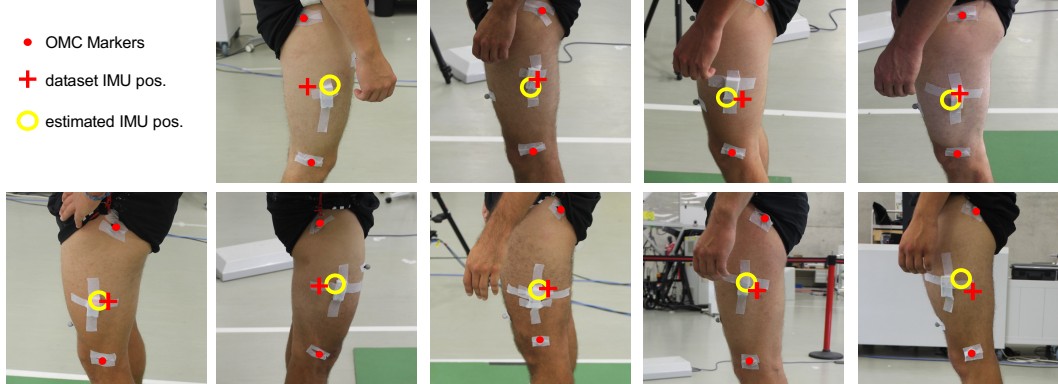

Figure 4: Comparison of IMU positionings from the dataset and our estimations. We use OMC markers as a reference frame. For all participants, we show either the right or left leg. We always chose the side where the IMU and OMC markers were clearly visible. If they were visible from both sides, we chose the picture that was taken more perpendicular to the sagittal plane.

A perfect simulation would require a correct multibody dynamics model with correct IMU positions. Our model and loss function can act together as a differential physical simulator. Therefore, we can optimize input parameters, including IMU orientations and positions. The IMU orientations and positions are prone to errors as they are placed and measured manually. Therefore, we finetuned the network and the IMU positions and orientations jointly for about 40 minutes per participant. In Figure 4, we show the results of the IMU positioning optimization for all participants' thigh IMUs. We use the trochanter and knee markers as reference for the hip and knee joints. We present the manually measured position of the thigh IMU in the dataset, which Dorschky et al. (2024) assumed was located on the segment axis. We show that we are able to recover this misplacement from the dataset. For most participant, the position estimation is on or very close to the IMU housing. To

our knowledge, current methods can only estimate the distance of an IMU from the joint center, but not the distance of the IMU to the segment axis. This discrepancy between the positioning from the dataset and our estimation could only be found for the thigh IMUs and that the margin of improvement in the metrics is very small (see C). However, the personalization of the IMU can make the model more robust to misplacements and misalignments then donning the IMUs. There is no validation for the correctness of body constants and ground contact model parameters on the given dataset, as that would require medical imaging. Thus, we excluded these parameters from the IMU positioning optimization. However, when we optimized the body constants, we found that only the moments of inertia yielded unrealistic values, as they converged to zero. We found these results because Kane's loss formulation favours smaller moments of inertia, as they lead to less forces and therefore less physics error in general. For the body weight, the same issue would apply, but we mitigated that by optimizing for the weight distribution instead of the body weight itself.

### 4.3 Sparse IMU Configurations

In practical use, the fewer IMUs one has to wear, the better. We have retrained the Bi-LSTM from scratch on configurations with only the foot-worn IMUs (F), foot and thigh IMUs (FT), and foot and pelvis IMUs (FP). Errors generally increased (Table 2), but the output motion is still physically and visually plausible (see C). For the running motions in F and FT configurations, the ankle angle and therefore the origin of the GRF is visibly shifted. Between the configurations with and without a pelvis IMU, the trunk orientation is different for all motions. Therefore, there is likely a discrepancy between the actual, physically plausible, trunk orientation and the IMU orientation, i.e. the pelvis IMU might not be correctly aligned. Compared to Dorschky et al. (2023), our increases in errors are similar for the F and FP configurations, but higher for the FT configuration. We believe our method is more affected by soft tissue artefacts, measurement errors caused by the movement of skin and muscle, from thigh IMUs compared to the optimal control method. As there is no hard constraint in SSPINNpose, it can trade off physical correctness for a better fit to the IMU signals, especially when they contain noise. On the other hand, optimal control's hard constraints not allowing physically incorrect motions. The pelvis and foot IMUs, on the other hand, are less affected by soft-tissue artefacts.

Table 2: Comparison of different sparse IMU configurations using the Bi-LSTM model on the evaluation metrics. The best results are shown in bold.

| IMU configuration | JAE [deg] | JTE [BWBH%] | GRF [BW%] | Speed [m s$^{-1}$] |
|---|---|---|---|---|
| All | **8.9** | 5.0 | **18.8** | **0.15** |
| Feet + Thighs | 14.4 | 8.1 | 32.7 | 0.45 |
| Feet + Pelvis | 12.6 | **4.9** | 24.9 | 0.41 |
| Feet | 13.2 | 7.4 | 27.8 | 0.30 |

## 5 Conclusion

In this work, we present SSPINNpose, a real-time method for the estimation of human movement dynamics from inertial sensor data that does not require labeled training data. Instead, it relies on self-supervision and physics information to find plausible motions. We show that SSPINNpose can accurately estimate joint angles, torques, and GRFs from IMU data, while outperforming state-of-the-art methods in terms of horizontal speed estimation. Additionally, SSPINNpose effectively identifies movement patterns from sparse IMU configurations and personalizes IMU placement on the body. Given its capability to work with minimal IMU configurations and allow for personalization, SSPINNpose is a promising approach for long-term monitoring of athletes and understanding injury mechanisms. In the future, we aim to extend SSPINNpose to 3D applications and adapt it for model predictive control tasks.

REPRODUCIBILITY STATEMENT

We provide the code for SSPINNpose in our supporting material and will link our github project in the final version. The addBiomechanics repository will also be linked in the final version.

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

## A  Implementation Details

**RNN & Hyperparameters:** We use a network architecture similar to physics inertial poser (PIP) (Yi et al., 2022). We use a LSTM with 2 layers with a hidden size of 256, while the output layers are of size 128 and 46, respectively. The LSTM has a dropout rate of $40\,\%$. Further hyperparameters, including the weighting between the loss terms, are listed in table 3. We take the hyperparameters from PIP, as we use the same architecture. The loss weights were tuned manually.

Table 3: Hyperparameters in SSPINNpose.

| Parameter | Value |
|---|---|
| learning rate | $10^{-3}$ |
| optimizer | Adam |
| batch size | 32 |
| criterion | MSE |
| $\eta_{imu}$ | 0.25 |
| $\lambda_K$ | 3.0 |
| $\lambda_T$ | 3.0 |
| $\lambda_{IMU}$ | 30.0 |
| $\lambda_{ankle}$ | 100.0 |
| $\lambda_B$ | 10000.0 |
| $\lambda_\tau$ | 1.0 |
| $\lambda_{slide}$ | 30.0 |
| $\lambda_{FS}$ | 1.0 |

**Calculation of point kinematics:** We list the equations to calculate the global kinematics, containing the positions $(x, y)$ and angle $\alpha$, of a point $\boldsymbol{p} = \{x, \dot{x}, \ddot{x}, y, \dot{y}, \ddot{y}, \alpha, \dot{\alpha}, \ddot{\alpha}\}$, based on its parent segment, here. For calculation, the parent is defined by an offset $d_x, d_y$, a point $\boldsymbol{p}' = \{x', \dot{x}', \ddot{x}', y', \dot{y}', \ddot{y}', \alpha', \dot{\alpha}', \ddot{\alpha}'\}$. First, $\{\alpha, \dot{\alpha}, \ddot{\alpha}\}$ are set by adding the local coordinates $\boldsymbol{q}_p$ to $p'$ for the respective point. Then, $\{x, \dot{x}, \ddot{x}, y, \dot{y}, \ddot{y}\}$ are calculated as follows:

$$x = x' + \cos(\alpha')d_x - \sin(\alpha')d_y, \tag{5}$$

$$y = y' + \sin(\alpha')d_x + \sin(\alpha')d_y, \tag{6}$$

$$\dot{x} = \dot{x}' - (\sin(\alpha')d_x + \cos(\alpha')d_y)\,\dot{\alpha}', \tag{7}$$

$$\dot{y} = \dot{y}' + \left( -\sin(\alpha')d_y + \cos(\alpha')d_x \right) \dot{\alpha}', \tag{8}$$

$$\ddot{x} = \ddot{x}' + \left( -d_x\dot{\alpha}'^2 - \ddot{\alpha}'d_y \right) \cos\alpha' - \left( -d_y\dot{\alpha}'^2 + \ddot{\alpha}'d_x \right) \sin\alpha', \tag{9}$$

$$\ddot{y} = \ddot{y}' + \left( -d_x\dot{\alpha}'^2 - \ddot{\alpha}'d_y \right) \sin\alpha' + \left( -d_y\dot{\alpha}'^2 + \ddot{\alpha}'d_x \right) \cos\alpha'. \tag{10}$$

The global kinematics are only directly estimated for the pelvis and the ankle. Therefore, the global kinematics based on the pelvis are first calculated for the hip joint position and pelvis IMU and then propagated along the kinematic chain. From the ankle kinematics that are seperately estimated, the heel and ankle point globals are calculated.

**Ground contact model:** We determine the ground contact point based on the global ankle rotation $\alpha_{ankle}$, where the contact point is positioned on the line between heel and toe. The exact position is determined as $(\tanh(\alpha_{ankle} * 7) + 1)/2$, where 1 corresponds to the toe and 0 to the heel. The GRF is calculated as: $\boldsymbol{F}_y = -k\zeta(\beta\boldsymbol{p}_{gc,y})\left(1 - b\dot{\boldsymbol{p}}_{gc,y}\right)/\beta$ with $\beta = 300$, stiffness $k = 100\,\mathrm{BW/m}$, damping $b = 0.75\,\mathrm{N\,s\,m^{-1}}$, and $\boldsymbol{F}_x = \mu_{max}\tanh(\hat{\mu})\boldsymbol{F}_y$, with $\mu_{max} = 0.5$. The global ankle kinematics $\tilde{\boldsymbol{p}}_{ankle}$ are estimated seperately and supervised by the estimated forward kinematics of the ankle $\boldsymbol{p}_{ankle}$:

$$\mathcal{L}_{GC} = \frac{1}{n_{ankle}} \sum_{i=1}^{n_{ankle}} \left( \left( \tilde{\boldsymbol{p}}_{ankle} - \boldsymbol{p}_{ankle} \right) / \sigma(\boldsymbol{p}_{ankle}) \right). \tag{11}$$

**Bounds on joint limits and maximum velocity:** For hip and ankle, we set the joint ranges to $[-\pi/3, \pi/3]$. As the knee can extend less, its joint range was set to $[-\pi/3, 0.1]$. The maximum velocity was set to $[-10\,\mathrm{m\,s^{-1}}, 10\,\mathrm{m\,s^{-1}}]$, while the vertical root position was set to $[0\,\mathrm{m}, 2\,\mathrm{m}]$.

**Equations for the auxiliary losses:** The torque minimization loss is calculated as:

$$\mathcal{L}_\tau = \sum \boldsymbol{\tau} / max(\dot{\boldsymbol{p}}_{0,x}, 1), \tag{12}$$

where $\boldsymbol{\tau}$ is the joint torque and $\dot{\boldsymbol{p}}_{0,x}$ is the speed of the root translation in the sagittal plane. The sliding penalty loss is calculated as:

$$\mathcal{L}_{slide} = \frac{1}{n_{gc}} \sum_{i=1}^{n_{gc}} \left( |\dot{\boldsymbol{p}}_{gc,x}|\boldsymbol{F}_{gc,y} \right), \tag{13}$$

where $\dot{\boldsymbol{p}}_{gc,x}$ is the horizontal speed of the foot and $\boldsymbol{F}_{gc,y}$ is the vertical GRF. The foot speed loss is calculated as:

$$\mathcal{L}_{FS} = \frac{1}{2} \sum_{\boldsymbol{p} \in \boldsymbol{p}_{ankle}} |\dot{\boldsymbol{p}}_x - \dot{\boldsymbol{p}}_{K,x}| - 0.3 max(\dot{\boldsymbol{p}}_{K,x}), \tag{14}$$

where $\dot{\boldsymbol{p}}_{K,x}$ is the reconstructed horizontal speed of the foot-worn IMU and $\dot{\boldsymbol{p}}_x$ is the estimated horizontal speed of the foot-worn IMU.

# B  COMPARISON TO 3D POSE ESTIMATION

Current state-of-the-art 3D pose estimation methods are typically evaluated on different metrics than those that biomechanists are interested in, which are listed in table 4: *1.) Jitter:* The third derivative of the joint positions in $\mathrm{km\,s^{-3}}$. *2.) Global Orientation Error (GOE):* The mean absolute error (MAE) between estimated global segment orientations and those obtained from addBiomechanics, including the root orientation, in degrees. This term is similar to the SIP error, which measures the accuracy of global limb orientations in 3D. *3.) Mean Absolute Joint Angle Error (JA-MAE):* The MAE between estimated joint angles and those obtained from addBiomechanics, including the root orientation, in degrees. *4. Joint Positioning Error (JPE):* The mean distance between the knee and ankle position in our estimation and the position of the respective OMC marker, in $\mathrm{cm}$. The greater trochanter marker was aligned with the hip joint in our estimations.

Compared to PIP (Yi et al., 2022), our results show lower angular errors, slightly higher positioning errors and higher jitter. None of these metrics is directly compareable due to different reasons:

Table 4: The top half shows results of our baseline models on more additional metrics for walking and all movements. For comparison, results from PIP (Yi et al., 2022) are listed in the bottom half on its datasets.

| SSPINNpose | Jitter [km s$^{-1}$] | GOE [deg] | JA-MAE [deg] | JPE [cm] | Latency (ms) [ms] |
|---|---|---|---|---|---|
| Walking | 0.75 | 4.9 | 6.7 | 6.8 | 3.5 |
| All motions | 1.95 | 6.9 | 7.0 | 6.5 | 3.5 |
| **PIP (Dataset)** | | **SIP** [deg] | | | |
| DIP-IMU | 0.24 | 15.02 | 8.73 | 5.04 | 16 |
| TotalCapture | 0.20 | 12.93 | 12.04 | 6.51 | 16 |

- Different model configuration: SMPL (Loper et al., 2015) is a 3D model, which PIP used, that contains more joints and rotational degrees of freedom. Therefore, the rotational errors can be bigger, while the joint positions are closer to the reference data. The positioning of joints and their distances to the aligned root joint also influences the metrics. Jitter is affected similiarly as JPE.

- Different evaluation method in JPE: In state-of-the-art methods, the reference joint centers are found by fitting SMPL to the reference data. On the other hand, we believe that the sagittal position of the knee and ankle markers is more precisely reflecting the actual joint position. By this, our error contains propagates inaccuracies in scaling the multibody dynamics model and thus reveals IMU-driven model personalization as a new challenge.

- The datasets are different. Besides walking, DIP-IMU and TotalCapture contain gestures, freestyle and range of motion movements. Therefore, there is no fair comparison between our method and PIP.

# C   ADDITIONAL RESULTS

**Physics Finetuning and personalization of IMU positions:**   We list the visual (see Figure 5) results of the physics finetuning and quantitative results (see Table 5) of the physics finetuning and IMU positioning personalization experiments. GRFs and joint torques are estimated more accurately, while the joint angles show slightly higher error.

Table 5: Quantitative comparison between the physics-finetuned model, personalized IMU orientations and rotation, and the baseline model.

| IMU configuration | JAE | JTE | GRF | Jitter | Speed |
|---|---|---|---|---|---|
| Baseline | **8.9** | 5.0 | 18.8 | 1.95 | 0.15 |
| Physics Finetuned | 9.3 | **4.5** | **14.9** | **1.15** | 0.20 |
| Personalized | 9.0 | 5.0 | 17.8 | 1.92 | **0.14** |

**Ablations:**   To justify the importance of the individual loss terms and implementation details, we performed an ablation study. The results are shown in Table 6. The ablations are explained as follows: *1.) w/o est-ankle:* We do not estimate ankle kinematics seperately, we use the full-body kinematics to estimate the GRFs instead. *2.) w/o input noise:* We remove the input noise from the IMU signals. *3.) w/o GRF minimum:* We remove the minimum bound on the GRFs. *4.) w/o $\mathcal{L}_{FS}$:* We remove the foot speed loss. *5.) w/ two contact points:* Instead of defining a single contact point based on the global foot angle, we set a fixed contact points for the foot and the heel, respectively. This is similar to the ground contact model in Dorschky et al. (2019).

We show that all ablations lead to a decrease in performance. We note that the GRF minimum is especially important because it prevents local minima where the model does not learn to interact with the ground.

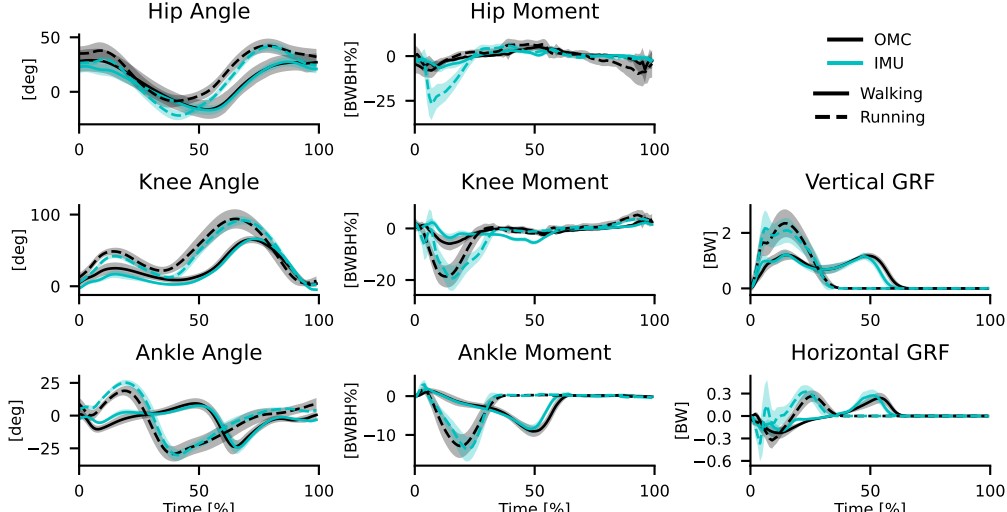

Figure 5: Average joint angles, torques and GRFs for the right leg, estimated with a physics-finetuned Bi-LSTM baseline model. We segmented the gait cycles during which the force plate was hit and normalized them to a duration of 100 samples. Walking and running data is shown in solid and dashed lines, respectively. Our estimations are shown in cyan, the reference data is shown in black. The shaded area represents the standard deviation.

Table 6: Quantitative results from the ablation study

| Model Version | JAE | JTE | GRF | Jitter | Speed |
|---|---|---|---|---|---|
| Full | **8.9** | 5.0 | 18.8 | **1.95** | **0.15** |
| w/o est-ankle | 9.4 | **4.7** | 27.1 | 3.59 | 0.20 |
| w/o noise augmentation ($\eta_{imu} = 0$) | 9.1 | 5.0 | **17.7** | 2.34 | **0.15** |
| w/o GRF minimum | 34.0 | - | - | 2.95 | 0.86 |
| w/o $\mathcal{L}_{FS}$ | 9.7 | 5.4 | 19.3 | 2.25 | 0.18 |
| w/ two contact points | 12.8 | **4.7** | 21.6 | 2.05 | 0.30 |

**SSPINNpose combined with PIP's second stage:** To combine PIP with SSPINNposes second stage, we first estimate the global kinematics with SSPINNpose and track the joint angles $q$ and joint velocities $\dot{p}$ with a PD controller. Instead of projecting contact polygons from contact points, we deviated from the original PIP implementation by using the heel and toe positions as the contact points. We heuristically set the ground contact probablity as a function of the horizontal speed of the contact points and estimated GRFs. The PD controller was tuned manually ($k_\lambda = 0.01$, $k_{res} = 0.1$, $k_\tau = 0.1$).

**Sparse IMU configurations:** In Figure 6, we show the stick figures for the sparse IMU configurations. We show that the model is able to estimate physically and visually plausible motions for all configurations. The errors are higher for the foot and thigh (FT) configuration, as the thigh IMUs are more prone to soft-tissue artefacts. The errors are lowest for the foot and pelvis (FP) configuration, as the pelvis IMU is less affected by soft-tissue artefacts.

**Sensitivity to IMU misplacement:** We retrained SSPINNpose with IMU positions that were perturbed by varying offsets in the sagittal plane. We did a training run each with offsets of $2\,\mathrm{cm}$, $4\,\mathrm{cm}$, $6\,\mathrm{cm}$, $8\,\mathrm{cm}$, $10\,\mathrm{cm}$, $15\,\mathrm{cm}$, and $20\,\mathrm{cm}$ in a random direction. Next, we tested whether SSPINNpose was able to recover either the IMU positions from the finetuning experiment in Section 4.2 or the IMU positions in the dataset. The results in Table 7 indicate that SSPINNpose is robust to small variations in IMU placement, but performance degrades with offsets of 10 cm or more. Manual IMU placement or measurement should usually be accurate within less than 10 cm, so we consider this a reasonable level of robustness. Recovering the IMU positions in the dataset is more challenging, but

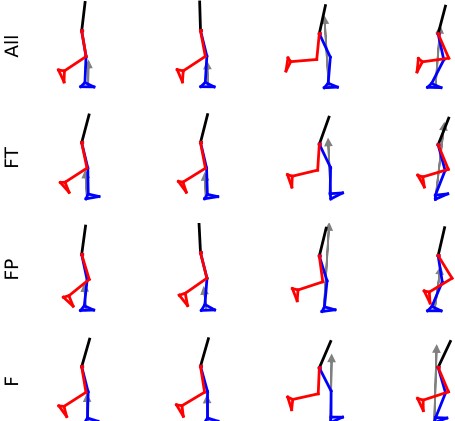

Figure 6: Sample stick figures for sparse IMU configurations, with forces annotated in gray. The rows show (from top to bottom) all IMUs, foot and thigh (FT) IMUs, and foot and pelvis (FP) IMUs, only foot (F) IMUs. We show random samples with the first two columns showing walking data, and the last two columns showing running data. All samples are drawn randomly from different participants.

as long the model has been trained on reasonably accurate IMU positions, it can partially recover the IMU positions. This is shown by the optimized offsets compared to the finetuning experiment and dataset being smaller than the offsets used for training.

Table 7: Experiments regarding the sensitivity to IMU misplacement and recovery of IMU positions. The first four columns show the evaluation metrics when training with offset IMU positions. The last two columns show the offsets compared to the original finetuning experiment the IMU positions in the dataset (DS).

| Offset | JAE | JTE | GRF | Speed | Offset vs. Sec. 4.2 | Offset vs. DS |
|---|---|---|---|---|---|---|
| cm | [deg] | [BWBH%] | [BW%] | [$\mathrm{m\,s^{-1}}$] | cm | cm |
| Baseline | 8.9 | 5.0 | 18.8 | 0.15 | 0.0 +- 0.0 | 4.7 +- 2.2 |
| 2 | 9.4 | 5.5 | 19.1 | **0.13** | 4.4 +- 2.0 | 2.2 +- 1.2 |
| 4 | 9.7 | 5.9 | 19.5 | 0.14 | 6.1 +- 2.5 | 4.3 +- 2.5 |
| 6 | **8.4** | 5.1 | **17.6** | **0.13** | 3.8 +- 2.3 | 2.1 +- 1.4 |
| 8 | 10.4 | 5.7 | 20.9 | 0.16 | 5.8 +- 2.3 | 4.1 +- 1.8 |
| 10 | 16.4 | **4.7** | 36.1 | 2.99 | 7.6 +- 3.3 | 6.3 +- 3.7 |
| 15 | 14.1 | 6.0 | 22.1 | 0.52 | 15.1 +- 8.7 | 14.5 +- 9.9 |
| 20 | 18.5 | 5.3 | 19.7 | 0.39 | 24.1 +- 11.6 | 23.7 +- 12.5 |

## D GRAPHICAL OVERVIEW OF SSPINNPOSE TRAINING AND EVALUATION SCHEME

In Figure 7, we give an overview of the training and evaluation scheme of SSPINNpose. The explaination to the graphic is as follows: **A:** We take continuous IMU signals, body constants, IMU positions and ground contact model parameters as input. **B:** We use a (Bi-) LSTM to output kinematics and joint torques. **C:** We show a stick figure of the estimated kinematics at $\{2.5, 3.0, ..., 4.5\}$ s. For two out of these frames, we also show the reference kinematics in grey. **D:** We supervise our model using the loss functions introduced in Section 3.2. Here we show: *1.) Kane's Loss*, which has the same dimensionality as the multibody dynamics model's degrees of freedom. *2.) Temporal Consistency Loss* for $\boldsymbol{p}_{ankle,r}$, where the estimated velocity is shown in black and the estimated acceleration in red. The dashed lines represent the numerical differentiation of the position and velocity, respectively. *3.) Virtual IMU:* The simulated IMU signals of a foot-worn IMU. *4.) Foot-IMU speed:* The estimated speed of the foot-worn IMU, our model in blue and the kalman-filter based

integration in green. The shaded area marks the zone where the speed error is zero. **E:** We show the biomechanical outcome variables. Dashed lines represent the reference data. *1.) Kinematics:* Hip flexion: blue; knee flexion: red; ankle plantarflexion: green. *2.) Speed:* Translational velocity. *3.) Torques:* Knee flexion: red, ankle plantarflexion: green. The hip flexion torque is not shown as it is out of range, but it is not estimated correctly for this trial. *4.) GRFs:* Vertical: blue, horizontal: red.

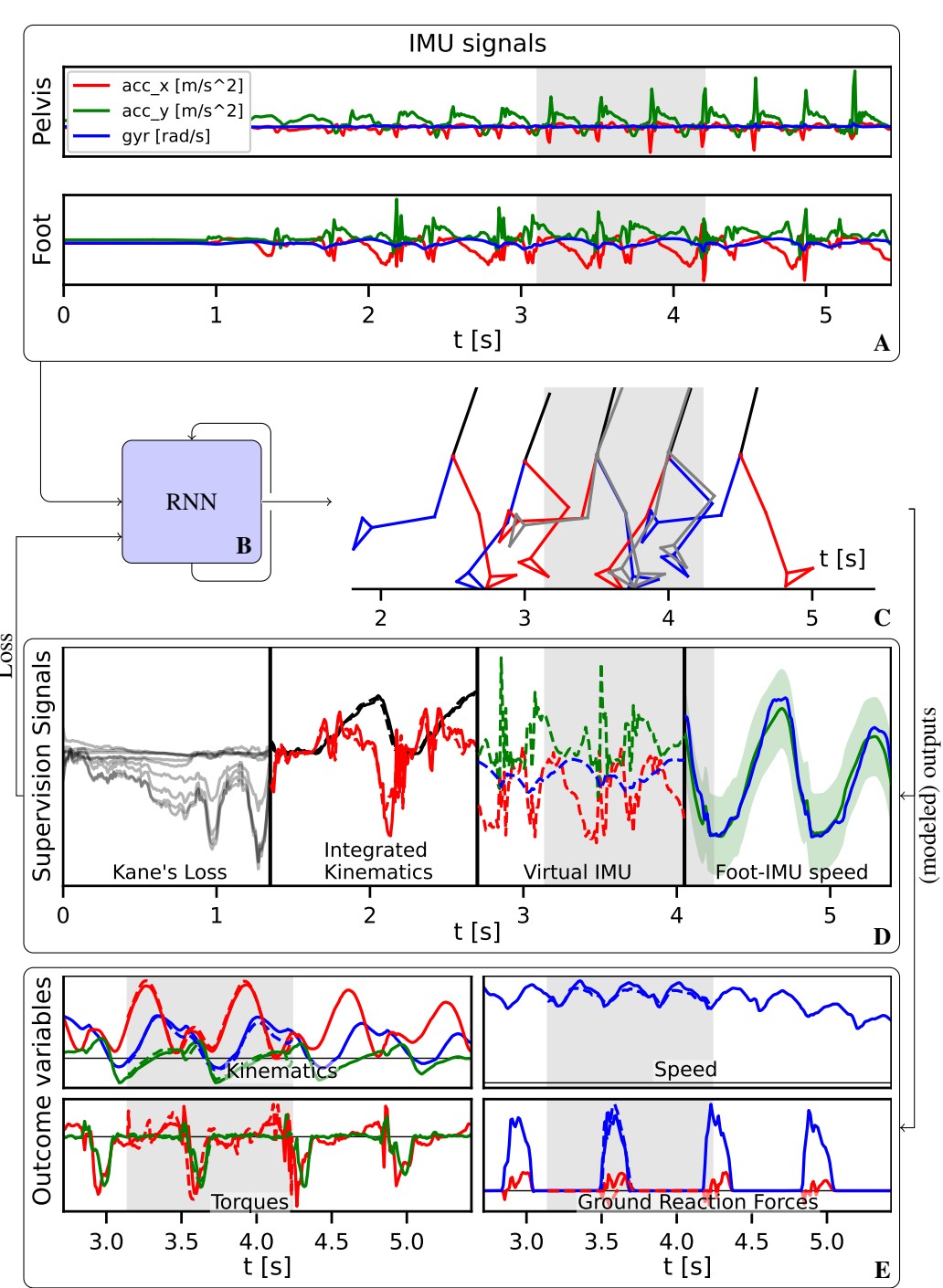

Figure 7: Overview of the SSPINNpose training and evaluation process. All data shown is from a single running bout at a max speed of $4.9\,\mathrm{m\,s^{-1}}$. The shaded area marks the time where the reference data was recorded.

