# OpenReview forum: "A Self-Supervised PINN for Inertial Pose and Dynamics Estimations"
_ICLR.cc/2025/Conference — Submitted to ICLR 2025_

### Official Review · Reviewer_Nikt · 2024-11-03

**Soundness:** 2
**Presentation:** 2
**Contribution:** 2
**Rating:** 5
**Confidence:** 4

**Summary:**

The paper proposes SSPINNpose, a self-supervised physics-informed neural network for estimating human movement dynamics using inertial measurement units (IMUs) for real-time applications in uncontrolled environments. SSPINNpose leverages a combination of physics-informed losses and recurrent neural network (RNN) architecture to infer both kinematic and kinetic properties of human motion.

**Strengths:**

- The paper estimates dynamics as a byproduct of kinematic alignment of actual IMU and estimations from a virtual model. I find this as the most interesting aspect of the paper.

- The authors use auxiliary physics-inspired losses for attempting realistic estimations.

**Weaknesses:**

> The authors highlight that most state-of-the-art methods rely on supervised approaches, while their method does not require labeled data, and that, this is one of the highlight of their work.

- However, as such, in this case of joint kinematics and kinetics regression, there is nothing as labeled or unlabeled data because the authors are anyway using full lower-body motion. The question of labeled or unlabeled data may appear in cases of classifying the intent such as walk level ground, climb stairs etc., but in the case where the goal is to regress body kinematics or kinetics, anyway, the recording full body motion is done, thus, there is no question of explicitly labeling the data.

- Thus, both supervised and self-supervised approaches appear to aim at the same objective: minimizing the deviation between experimental and estimated kinematics. So here, the self-supervised goal seems to be only a wording and framing problem.
It is also evidenced by the main self-supervised objective that the authors use, minimizing the deviation between actual IMU motion and the virtual model’s motion, which closely resembles a supervised objective. Could the authors clarify how their approach differs fundamentally from traditional supervised methods? Further, the fact that dynamics is estimated as a byproduct of minimizing kinematic deviations could be done by posing this as a supervised method also, since the authors are estimating GRF using a ground contact model anyway.

> I found the objectives, contributions, and methodology of this work weakly positioned and unclear, as the authors highlight multiple objectives to establish the novelty of their method over existing approaches. Could the authors to clarify the primary advantages of their approach and how it complements or differs from existing state-of-the-art tools. Here are some specific points of feedback:

- One stated objective appears to be the estimation of kinematics and kinetics using IMUs rather than optical motion capture systems. Then why go for a self-supervised method, which may be less accurate? Musculoskeletal models, e.g., OpenSim can do this too. AFAIK, OpenSim 4.1 has begun supporting IMU data analysis. Could the authors elaborate on the specific advantages of their self-supervised approach compared to these methods? It appears that the real-time estimation is the only advantage. Then why this highlight of IMUs vs. OMC?

- Another goal mentioned is to facilitate analyses outside laboratory settings. This could theoretically be achieved by collecting data in real-world environments and applying OpenSim or similar tools to calculate inverse kinematics and dynamics. Could the authors clarify the added value of SSPINNpose for these applications, particularly if it avoids preprocessing requirements that are still necessary for lab-based systems?

- The authors also emphasize real-time computation of kinematics and kinetics. This seems particularly relevant to their approach, as OpenSim and related tools typically do not offer real-time capabilities. However, did the authors test this model in real-time by actually doing lower-limb gait analysis in the wild? Could the authors describe the setting in detail and elaborate on the real-time metrics like what was the real-time prediction frequency, lost packets, how the sensor synchronization and failure were handled, read rate etc? Sorry if I missed the information.

> The authors miss many common regression-based works that estimate joint kinematics regression of one joint from other inputs such as [1], [2], [3], [4], etc.
 - Many such works are missing in the related works.
 - Comparison with many such common existing methods also seem to be missing.

> When posed as a self-supervised problem, how do the authors handle collapse, especially, when the virtual model estimation is so perfect that the difference between the actual and virtual model estimations are almost trivial?

> Do the authors evaluate other similar self-supervised methods like [5] on their data?




References:

[1] Lim, Hyerim, Bumjoon Kim, and Sukyung Park. "Prediction of lower limb kinetics and kinematics during walking by a single IMU on the lower back using machine learning." Sensors 20.1 (2019): 130.

[2] Lower body kinematics estimation from wearable sensors for walking and running: A deep learning approach. Gait & posture, 83, pp.185-193.

[3] A function approximator model for robust online foot angle trajectory prediction using a single IMU sensor: Implication for controlling active prosthetic feet. IEEE Transactions on Industrial Informatics, 2022

[4] Estimation of Lower Limb Joint Angles and Joint Moments during Different Locomotive Activities Using the Inertial Measurement Units and a Hybrid Deep Learning Model. Sensors 23.22 (2023): 9039.

[5] Large-scale Training of Foundation Models for Wearable Biosignals, ICLR 2024

**Questions:**

Please see the weaknesses section

---

> ### Author Response · Authors · 2024-11-18
> **Response to Reviewer Nikt, Part 1**
>
> Dear Reviewer Nikt,
>
> thank you for your feedback and suggestions. Currently, we are preparing additional experiments and will provide a revised version of the paper when finished. We will address your comments in the following way in the revised version:
>
> ## Weaknesses:
>
> > ... However, as such, in this case of joint kinematics and kinetics regression, there is nothing as labeled or unlabeled data because the authors are anyway using full lower-body motion. ...
>
> We disagree with your statement. By "labels", we refer to the ground truth motion being known, which usually means that the motions are recorded with a reference system (e.g. optical motion capture). We apologize for the confusion and will clarify this in the revised version.
>
> > Thus, both supervised and self-supervised approaches appear to aim at the same objective: minimizing the deviation between experimental and estimated kinematics. So here, the self-supervised goal seems to be only a wording and framing problem. ...
>
> The main goal of both supervised and self-supervised approaches is to minimize the deviation between experimental and estimated kinematics. However, in the supervised case, the recorded ground truth motions are used in training. In the self-supervised case, the ground truth motion is unknown and the model has to find it from the IMU data alone. In practice, the difference is that for every motion that a supervised model should be able to predict, that motion has to be recorded with a reference system, which is time-consuming and typically limits the motions to lab environments. For the self-supervised model, the model can learn the motion from IMU data only, which can easily be recorded in any environment.
>
> > ... Further, the fact that dynamics is estimated as a byproduct of minimizing kinematic deviations could be done by posing this as a supervised method also, since the authors are estimating GRF using a ground contact model anyway.
>
> Dynamics are not a byproduct of minimizing kinematic deviations. Minimizing $\mathcal{L}_{\text{IMU}}$ alone can be ambiguous, especially in sparse IMU setups. Therefore, we made the assumption that a physically plausible motion is more likely to be correct. And in order to be physically plausible, the dynamics have to be correct. Therefore, the method would not work without the dynamics. The ground reaction forces are part of the dynamics - we chose to model them with a ground contact model based on the kinematics that the LSTM predicts. This allows the network to learn physics-based foot-ground interactions.
>
> > One stated objective appears to be the estimation of kinematics and kinetics using IMUs rather than optical motion capture systems. Then why go for a self-supervised method, which may be less accurate? Musculoskeletal models, e.g., OpenSim can do this too. AFAIK, OpenSim 4.1 has begun supporting IMU data analysis. Could the authors elaborate on the specific advantages of their self-supervised approach compared to these methods? It appears that the real-time estimation is the only advantage. Then why this highlight of IMUs vs. OMC?
>
> The current gold-standard for estimating kinematics and kinetics is optical motion capture systems. However, IMUs have their advantages in being cheaper and more portable, which is why we motivated using IMUs in the introduction. OpenSim's IMU inverse kinematics solver (OpenSense) only allows for the estimation of joint angles, not joint torques. For the estimation of joint torques, force information would be required. That can either happen trough force plates or through an optimization as in [1]. The latter is not real-time capable and requires additional preprocessing steps. Our method only requires aligning the IMU data.
>
> [1] Karatsidis et al. "Musculoskeletal model-based inverse dynamic analysis under ambulatory conditions using inertial motion capture." Medical Engineering and Physics, 65, 2019.
>
> > The authors also emphasize real-time computation of kinematics and kinetics. This seems particularly relevant to their approach, as OpenSim and related tools typically do not offer real-time capabilities. However, did the authors test this model in real-time by actually doing lower-limb gait analysis in the wild? Could the authors describe the setting in detail and elaborate on the real-time metrics like what was the real-time prediction frequency, lost packets, how the sensor synchronization and failure were handled, read rate etc? Sorry if I missed the information.
>
> Thank you for the suggestion. We did not test a complete IMU system in the wild, but there are existing systems such as Xsens. Our claim is that our method can handle inference in real-time.

---

> ### Author Response · Authors · 2024-11-18
> **Response to Reviewer Nikt, Part 2**
>
> > The authors miss many common regression-based works that estimate joint kinematics regression of one joint from other inputs such as [1], [2], [3], [4], etc.
>
> Thank you for the suggestion. We have overlooked these works and will add them in the final version. We are especially happy for source [4] and are currently investigating the dataset they have used.
>
> [4] Estimation of Lower Limb Joint Angles and Joint Moments during Different Locomotive Activities Using the Inertial Measurement Units and a Hybrid Deep Learning Model. Sensors 23.22 (2023): 9039.
>
> > When posed as a self-supervised problem, how do the authors handle collapse, especially, when the virtual model estimation is so perfect that the difference between the actual and virtual model estimations are almost trivial?
>
> We have not observed any collapsing behavior in our experiments.
>
> > Do the authors evaluate other similar self-supervised methods like [5] on their data?
>
> Thank you for the suggestion. It will be very interesting what [5] can achieve with IMU data. Testing that, however, is out of the scope of this paper: First, the dataset in [5] is some orders of magnitudes larger, and secondly, while it will be possible that kinematics and dynamics would be encoded in the foundation model, it is unclear to us how to extract this information without labeled data.
>
> [5] Large-scale Training of Foundation Models for Wearable Biosignals, ICLR 2024

---

> ### Comment · Reviewer_Nikt · 2024-11-24
>
> Thank you for your response.
>
> I find it difficult to understand the distinction the authors make between "ground truth" and "no ground truth." The claim that motion recorded via a MOCAP system constitutes ground truth, whereas motion recorded via IMUs does not, is inaccurate. These are simply two different methods of capturing the same data, i.e., full-body kinematics. In Mocap experiments, we can additionally record Ground Reaction Forces (GRFs) using force plates, which can also be done in experiments coupling with IMUs. These GRFs along with the kinematics can be used for estimating the dynamics, such as, joint torques. If the paper aims to predict kinematics from IMU data using IMU data as input, their ground truth is the IMU data itself and this would fall under the category of supervised learning. Based on the problem formulation and the results presented (e.g., Figure 3), it seems that the estimation of kinematics is also being positioned as a contribution. The problem formulation (lines 148–149) states:
> "*The outputs are the kinematics of the lower body, including root rotation and translations, joint angles,* and joint torques." In this context, the self-supervised formulation appears misaligned with the methodology.
>
> As I mentioned in my review, estimating dynamics from kinematics may arguably be considered as not having ground truth, and I noted this as the most interesting aspect of the paper, however, as the paper is positioned now, this constitutes only one part of the paper.
>
> As someone from this field, I observe several inaccuracies in the paper's focus and formulation that require correction. I would strongly suggest that the authors reposition the paper more accurately based on the given comments.
>
> I would therefore maintain my score.

---

> > ### Author Response · Authors · 2024-11-25
> > **Response to Reviewer Nikt**
> >
> > Dear Reviewer Nikt,
> >
> > Thank you for your thoughtful comments. We appreciate the opportunity to clarify our methodology and address potential misunderstandings.
> >
> > We agree with your assessment that a MoCap or IMU system does not directly constitute ground truth kinematics. However, MoCap is widely considered the gold standard for estimating kinematics. Many existing datasets, such as TotalCapture [1] and AMASS [2], rely on MoCap systems for this purpose. In current deep inertial pose estimation methods, MoCap data from [1,2] are typically used as ground truth for training, with _raw_ IMU data being used as inputs. DIP-IMU [3] is an exception, where a _full-body_ IMU setup is used to provide training kinematics.
> >
> > In contrast, IMU data be used to estimate kinematics via Kalman Filters, but these approaches typically require a _full-body_ IMU setup to resolve ambiguities (Xsens) or impose additional constraints [4]. Our approach is distinct:
> >
> > - Our model is trained solely on _raw_ IMU data (acceleration and gyroscope signals), without requiring pre-computed kinematics as targets.
> > - It handles _sparse_ IMU setups, where kinematics are inherently ambiguous, setting it apart from methods requiring full-body IMUs for inference or MoCap/full-body IMU reference data for training.
> > - The network outputs both kinematics and dynamics, using dynamics to regularize kinematics during training.
> >
> > As our model is supervised by its own input data, we consider it a self-supervised method. We are aware that there are multiple definitions of self-supervised learning and that our approach does not align with all definitions. We use the self-supervised terminology to emphasize that our model does not require external labels, and distinguish it from a hypothetical supervised version of our model that uses PINN losses with MoCap or full-body derived IMU kinematics/dynamics as targets.
> >
> > [1] Trumble et al., "Total capture: 3d human pose estimation fusing video and inertial sensors.", BMVC, 2017.
> >
> > [2] Mahmood et al., "AMASS: Archive of Motion Capture as Surface Shapes", ICCV, 2019.
> >
> > [3] Huang et al., "Deeep Inertial Poser: Learning to Reconstruct Human Pose from Sparse Inertial Measurements in Real Time", ACM Transactions on Graphics, 2018.
> >
> > [4] van Marcard et al., "Sparse Inertial Poser: Automatic 3D Human Pose Estimation from Sparse IMUs", EUROGRAPHICS, 2017.

---

> > > ### Comment · Reviewer_Nikt · 2024-11-25
> > >
> > > Thanks, but the answer you provided does not relate to my comment or answer my comment. Could you please clarify your point or explain what you aim to address and how it connects to the discussion?

---

> > > > ### Author Response · Authors · 2024-11-25
> > > > **Response to Reviewer Nikt**
> > > >
> > > > We are trying to determine whether our approach can be classified as self-supervised learning and wanted to ensure there’s no mutual misunderstanding. We would appreciate your clarification on the following points:
> > > >
> > > > > If the paper aims to predict kinematics from IMU data using IMU data as input, their ground truth is the IMU data itself, and this would fall under the category of supervised learning.
> > > >
> > > > If we aim to predict kinematics from IMU data, and the ground truth is IMU data itself, how would we be able to supervise the learning of kinematics, given that there are no explicit labels for them? Could you elaborate on how this aligns with supervised learning in this context?
> > > >
> > > > > The outputs are the kinematics of the lower body, including root rotation and translations, joint angles, and joint torques. In this context, the self-supervised formulation appears misaligned with the methodology.
> > > >
> > > > None of these outputs in our approach have explicit corresponding labels. Based on this, how would you categorize our method (e.g., supervised, self-supervised, semi-supervised, or unsupervised)?
> > > >
> > > > We look forward to your insights, so that we can correctly position the methodology.

---

> > > > > ### Author Response · Authors · 2024-11-29
> > > > > **Clarification / Reminder**
> > > > >
> > > > > Dear Reviewer Nikt,
> > > > >
> > > > > In our revised manuscript, we have added a new section on related literature that highlights similar approaches from other domains. A relevant example that aligns with our approach at a high level is [1], which follows a similar structure: taking an input, estimating a value of interest, simulating the input modality, and computing the loss between the two. To illustrate:
> > > > >
> > > > > - [1]: Depth Maps → Estimate Kinematics → Render Depth Map → Loss
> > > > > - Ours: Accelerometer & Gyroscope Readings → Estimate Kinematics & its 1st and 2nd derivatives → Simulate Acc/Gyro → Loss
> > > > >
> > > > > We also kindly request your response to our previous questions to help resolve all outstanding points.
> > > > >
> > > > > Thank you for your time and consideration.
> > > > >
> > > > > [1] Wan et al. "Self-Supervised 3D Hand Pose Estimation Through Training by Fitting." CVPR, 2019

---

### Official Review · Reviewer_CE42 · 2024-11-04

**Soundness:** 3
**Presentation:** 3
**Contribution:** 3
**Rating:** 5
**Confidence:** 3

**Summary:**

The paper introduces SSPINNpose, a self-supervised physics-informed neural network designed to estimate human movement dynamics from inertial measurement unit (IMU) data without requiring ground truth labels. By integrating a physics model of the human body and leveraging virtual sensor data, the method enforces physical plausibility using Kane's equations and a temporal consistency loss. SSPINNpose is capable of accurately estimating joint angles and joint moments in real-time, even with sparse sensor configurations, and can infer the positions of the sensors on the body. Experiments demonstrate that SSPINNpose achieves comparable performance to supervised methods in estimating lower-body dynamics during walking and running at various speeds.

**Strengths:**

Innovation: Proposes a novel self-supervised approach to estimate human movement dynamics from IMU data without the need for ground truth labels, addressing a significant limitation in current methods.
Real-time Capability: Demonstrates that SSPINNpose can estimate dynamics in real-time with low latency, which is crucial for practical applications in clinical and sports settings.
Physical Plausibility: Incorporates physics-based constraints using Kane's equations and temporal consistency to ensure physically plausible motion estimations.
Sparse Sensor Configurations: Shows the method's versatility by effectively handling sparse IMU setups and inferring sensor placements, enhancing its practicality.

**Weaknesses:**

Limited to 2D Movements: The method focuses on lower-body dynamics in the sagittal plane and does not address full 3D motion estimation, limiting its applicability to more complex movements.
Simplifying Assumptions: Assumes flat ground and non-sliding feet, which may not hold in real-world scenarios and could affect the accuracy of estimations in such conditions.
Model Personalization: Uses a generic multibody dynamics model without personalization, potentially affecting the accuracy due to individual anatomical differences.
Replication of Noise: The model may replicate noise in IMU signals, which could reduce the physical plausibility of the estimated motions.

**Questions:**

How does SSPINNpose perform when applied to movements involving significant out-of-plane dynamics or complex 3D motions?
Can the method be adapted or extended to handle 3D movement estimation, and what challenges would that entail?
How sensitive is SSPINNpose to variations in IMU placement, and how robust is the sensor placement inference in practice?
What impact do the simplifying assumptions (e.g., flat ground, non-sliding feet) have on the method's performance in more variable real-world environments?
How does the model handle or mitigate the replication of noise in IMU signals to maintain physical plausibility?

---

> ### Author Response · Authors · 2024-11-18
> **Response to Reviewer CE42**
>
> Dear Reviewer CE42,
>
> Thanks for your feedback and interesting questions. Currently, we are preparing additional experiments and will provide a revised version of the paper when finished. We will address your comments in the following way in the revised version:
>
> ## Weaknesses:
> > Limited to 2D Movements: The method focuses on lower-body dynamics in the sagittal plane and does not address full 3D motion estimation, limiting its applicability to more complex movements.
>
> [same answer as to ig7e] We agree with you that a self-supervised 3D pose estimation method would be very beneficial. However, as the conceptually very similar optimal control problems have not been successfully solved for the 3D case, we decided to focus on the 2D case. Further challenges could be a significantly longer training time and possible higher chance of local minima. We leave that as an open challenge for future work.
>
> > Simplifying Assumptions: Assumes flat ground and non-sliding feet, which may not hold in real-world scenarios and could affect the accuracy of estimations in such conditions.
>
> In many activites of daily living, this assumption is valid. In future work, we want to investigate how to extend the method to also estimate the ground plane self-supervisedly, for example to investigate stair climbing.
>
> > Model Personalization: Uses a generic multibody dynamics model without personalization, potentially affecting the accuracy due to individual anatomical differences.
>
> The multibody dynamics model is in fact not personalized other than adjusting the height of participants. However, our method can be used to personalize these models - see lines 463-467 where we mention that we did do that, but could not evaluate it due to the lack of ground truth data. Personalization, especially when IMUs are the data source, is a largely unsolved problem in biomechanics, and we believe that our method is a step towards solving it.
>
> > Replication of Noise: The model may replicate noise in IMU signals, which could reduce the physical plausibility of the estimated motions.
>
> This is a valid concern. We show in the ablation study that the by augmenting the input data with noise, the model can learn to ignore the noise, thus making the output motion less jittery. A potential future direction is to develop a probabilistic version of the method, which would allow the model to learn the noise distribution and generate more realistic outputs. A higher weight on the physics terms can also reduce the impact of noise (see Appendix D). If Kane's loss and the physical consistency loss would reach zero, only the torque would be able to introduce jitter, however, we also regularize the torque.
>
> ## Questions:
> > How does SSPINNpose perform when applied to movements involving significant out-of-plane dynamics or complex 3D motions?
>
> We assume poorly. We are searching for failure cases.
>
> > Can the method be adapted or extended to handle 3D movement estimation, and what challenges would that entail?
>
> See answer to weaknesses above.
>
> > How sensitive is SSPINNpose to variations in IMU placement, and how robust is the sensor placement inference in practice?
>
> We are investigating this in current experiments. Thank you for the suggestion, that is very interesting to us.
>
> > What impact do the simplifying assumptions (e.g., flat ground, non-sliding feet) have on the method's performance in more variable real-world environments?
>
> Whenever simplifying assumptions do not hold, for example a significant slope or skipping on one foot, the model will likely fail. In these cases, different assumptions might be necessary. We leave this investigation for future work.
>
> > How does the model handle or mitigate the replication of noise in IMU signals to maintain physical plausibility?
>
> See answer to weaknesses above.

---

> > ### Author Response · Authors · 2024-11-25
> > **Update #1**
> >
> > > How does SSPINNpose perform when applied to movements involving significant out-of-plane dynamics or complex 3D motions?
> >
> > We uploaded new supplementary material where we show how SSPINNpose performs on out-of-plane and 3D motions with the examples of 180 degree jumps and lateral lunges. (supplementary/molinaro2024/out_of_distribution)

---

### Official Review · Reviewer_SLE6 · 2024-11-04

**Soundness:** 3
**Presentation:** 3
**Contribution:** 2
**Rating:** 6
**Confidence:** 5

**Summary:**

The paper introduces SSPINNpose, a self-supervised approach for estimating gait kinematics only from IMU data. The authors proposed a self-supervised approach to train a recurrent model based on LSTMs. The model is trained in an IMU signal reconstruction task without any labels while regularized by a multitude of physical constraints. Their approach demonstrates promising results in estimating joint angles, torques, and GRFs from IMU data, and the source code for testing/training is provided.

**Strengths:**

- The paper is well-written and easy to follow.
- The paper shows the potential of combining physics-informed objectives during self-supervision for gait kinematic estimation.

**Weaknesses:**

- **Lack of Comparisons**: One of the main weaknesses of the paper is its lack of direct comparisons with prior approaches on the same datasets and under the same conditions and evaluation criteria. I find the comparisons in the supplementary materials insufficient for judging the contribution and effectiveness of their work, especially given that the proposed method is not evaluated under the same conditions as prior works. Please see my suggestions below to address this limitation.
- **Contributions**: The paper proposes several physics constraints to guide the model's training. However, it is unclear what their contributions are in this combination. I have found several other optimization-based studies using the same physical constraints during pose/motion/gait kinematic estimation. Despite this, the paper's approach of combining physics-informed objectives during model training is somewhat unique. However, there are no experiments that support or highlight the impact of this novelty.
- **Evaluations**: The authors evaluate their approach only on one dataset, which contains only walking and running sequences. Therefore, the impact of this paper is only on some applications and its influence on others is not fully studied.

**Questions:**

The paper tackles an important problem in the pose and gait estimation fields and shows promising results in the same range as prior works on other datasets. However, I have some concerns regarding the evaluations and the lack of comparisons. Therefore, I suggest rejecting this paper in its current form. To improve the paper, I have several questions and suggestions:
1. Can you please provide direct comparisons with state-of-the-art methods on the same dataset and under the same evaluation conditions? In the first paragraph of page 8, the paper mentions that the dataset owners have used another evaluation scheme, and therefore the results are not directly comparable with the authors. To address this issue, my suggestion is to evaluate your approach with both schemes to back up your claims.
2. Can you please elaborate on the paper's contributions, beyond combining and tuning existing losses during self-supervised training? I suggest authors add a recap of their contributions to the last paragraph of the introduction to clarify and highlight their contributions.
3. Can you please evaluate the computational cost of adding all objective functions during training? Does it increase the training time?
4. Can you please elaborate on the difference between your approach and two-stage solutions, similar to Physical Inertial Poser (PIP)? I would be interested in knowing if there are any merits in a second-stage optimization after you have the initial results. In this case, some physics constraints like GRF may be necessary during the first stage (self-supervised learning) based on Table 8. But how necessary would the other losses be? One advantage of the second stage is that it can generalize well to unseen actions, which I believe your approach lacks. It would be valuable if the authors could provide more details on the advantages and disadvantages of their design vs optimization-based and hybrid (2-stage) solutions.


Some things could be improved, but did not impact the score:
- On some places, like L236 and L268, the authors say "see supplementary A/B/C", but on others, like L377, L382, and L359, they just say "See B/C/D". Please use a more constant formatting.
- On L240, "This loss is applied generalized coordinates q" seems like an incomplete sentence.
- The authors have used a simple LSTM network that is likely to have been fully tuned in prior works. It would be interesting to see a comparison with more modern signal processing architectures or a comparison with CNN approaches from prior works. What is the intuition behind choosing this network? Tables 5,6 and 8 are more informative for belonging to the main paper, while Table 1 is just a network comparison that should be supplementary.

---

> ### Author Response · Authors · 2024-11-18
> **Response to Reviewer SLE6, Part 1**
>
> Dear Reviewer SLE6,
>
> Thank you for your detailed feedback. We appreciate your suggestions for improvement. Currently, we are preparing additional experiments and will provide a revised version of the paper when finished. We will address your comments in the following way in the revised version:
>
> ## Weaknesses:
> > **Contributions:** The paper proposes several physics constraints to guide the model's training. However, it is unclear what their contributions are in this combination. I have found several other optimization-based studies using the same physical constraints during pose/motion/gait kinematic estimation. Despite this, the paper's approach of combining physics-informed objectives during model training is somewhat unique. However, there are no experiments that support or highlight the impact of this novelty.
>
> Thanks for pointing out this weakness. The main contribution of this paper is to convert these optimization-based approaches into a self-supervised learning method. In practice, the impact of this novelty is that it overcomes a key limitation of each current deep learning and optimization-based approaches: the need for labeled data and inference time.
> - Labeled data in deep learning: A new movement that is not in the dataset, e.g. a long jump, would require recording a new dataset containing that movement and reference data, which is limiting. In our approach, the model can learn the new movement from the IMU data alone, without the need for labeled data.
>  - Inference time in optimization-based approaches: Optimization-based approaches require a lot of time to compute the optimal solution. For example, similar 2D problems require ca. 30 mins for a single gait cycle [1]. These methods also scale poorly with estimated sequence length. Our method, on the other hand, can estimate the movement in real-time, scaling linearly. Even when you consider the training time, our method is considerably faster than optimization-based approaches when analyzing more data.
>
> [1] Dorschky, E., et al. "Comparing sparse inertial sensor setups for sagittal-plane
> walking and running reconstructions", bioRxiv, 2023.
>
> > **Evaluations:** The authors evaluate their approach only on one dataset, which contains only walking and running sequences. Therefore, the impact of this paper is only on some applications and its influence on others is not fully studied.
>
> - This is correct. In the sagittal plane, walking and running are the main movement types. For others (e.g. jumping, skipping on one foot, etc.), there are no suitable datasets available. Reviewer Nikt has pointed us to [2], which we are currently evaluating additionally.
>
> [2] Camargo, J. et al. "A comprehensive, open-source dataset of lower limb biomechanics in multiple conditions of stairs, ramps, and level-ground ambulation and transitions." Journal of Biomechanics, 2021.

---

> ### Author Response · Authors · 2024-11-18
> **Response to Reviewer SLE6, Part 2**
>
> ## Questions:
> > Can you please provide direct comparisons with state-of-the-art methods on the same dataset and under the same evaluation conditions? In the first paragraph of page 8, the paper mentions that the dataset owners have used another evaluation scheme, and therefore the results are not directly comparable with the authors. To address this issue, my suggestion is to evaluate your approach with both schemes to back up your claims.
>
> We agree that the evaluation is not directly comparable to the dataset owners. We will add a comparison to the dataset owners' evaluation scheme.
> - Comparing ours to [3]: We train and evaluate on ensemble averaged gait cycles per condition.
> - Comparing ours to [4]: We train on subjects 4-10 and evaluate on ensemble averaged gait cycles per condition for subjects 1-3.
>
> [3] Dorschky et al. "Estimation of gait kinematics and kinetics from inertial sensor data using optimal control of musculoskeletal models." Journal of Biomechanics, 2019.
>
> [4] Dorschky et al. "CNN-Based Estimation of Sagittal Plane Walking and Running Biomechanics From Measured and Simulated Inertial Sensor Data", Frontiers in Bioengineering and Biotechnology, 2020.
>
> > Can you please elaborate on the paper's contributions, beyond combining and tuning existing losses during self-supervised training? I suggest authors add a recap of their contributions to the last paragraph of the introduction to clarify and highlight their contributions.
>
> Thank you for the suggestion. We will add a recap of our contributions to the last paragraph of the introduction.
>
> > Can you please evaluate the computational cost of adding all objective functions during training? Does it increase the training time?
>
> The relative computation time of each objective and inference step, averaged over 100 batches, is listed here:
>
> | Objective / Step | Time (%)  |
> |------------------|-----------|
> | NN: Forward pass | 2.01% +- 0.33% |
> | Calculation of global coordinates | 69.98% +- 2.59% |
> | GRF estimation | 2.31% +- 0.41% |
> | $\mathcal{L}_{\text{GC}}$   | 18.56% +- 2.16% |
> | $\mathcal{L}_{\text{K}}$  | 4.26% +- 0.59% |
> | $\mathcal{L}_{\text{T}}$   | 0.84% +- 0.17% |
> | $\mathcal{L}_{\text{IMU}}$   | 0.03% +- 0.01% |
> | $\mathcal{L}_{\text{B}}$  | 0.97% +- 0.21% |
> | $\mathcal{L}_{\text{slide}}$ | 0.49% +- 0.11% |
> | $\mathcal{L}_{\text{FS}}$  | 0.31% +- 0.07% |
> | $\mathcal{L}_{\text{tao}}$  | 0.24% +- 0.05% |
>
> As seen above, the main biggest cost is the calculation of global coordinates. The auxiliary losses are negligible compared to the main losses. For context, $\mathcal{L} _ {\text{K}}$ , $\mathcal{L} _ {\text{IMU}}$, $\mathcal{L} _ {\text{FS}}$, $\mathcal{L} _ {\text{GC}}$, and the GRF estimation, make use of the global coordinates. The GRF estimation is then used in $\mathcal{L} _ {\text{K}}$, $\mathcal{L} _ {\text{B}}$, and $\mathcal{L} _ {\text{slide}}$.
>
>
> > Can you please elaborate on the difference between your approach and two-stage solutions, similar to Physical Inertial Poser (PIP)? ...
>
> Thank you for the suggestion, the comparison to PIP is indeed very interesting. We will add PIP to our method to evaluate the impact of PIP's second stage PD controller. It has to be noted that the SIP error in PIP (Table 2 in their paper) did not see much impact from the physics module or the PD controller, while the jitter was reduced substancially. We expect to see similar results in our method. We will elaborate on the advantages and disadvantages of their design vs optimization-based and hybrid (2-stage) solutions.
>
> > Some things could be improved, but did not impact the score: ...
>
> Thank you for the suggestions. We will correct these errors.
>
> > The authors have used a simple LSTM network that is likely to have been fully tuned in prior works. ... what is the intuition behind choosing this network? ...
>
>  - Intuition behind LSTM vs Transformer/CNN: We are generally interested in any universal function approximator, as our method is mainly concerned with the self-supervised learning of movement dynamics. Therefore, any network that can capture temporal dependencies is suitable. In our case, the hidden state of RNNs are advantageous: The temporal consistency loss states that there is a connection between y(t) and y(t+1), therefore, we believe that having information about y(t-1) during the estimation of y(t) is beneficial. For transformers/CNNs, we would need to give them that information explicitly, as they would essentially operate on shifting windows. Furthermore, previous work [DIP, TransPose, PIP] also made use of LSTMs.
>  - Tables: Thank you for the suggestion. We will rearrange the tables in the final version, depending on page limits.
>
> Edits: LaTeX formatting in comment

---

> ### Author Response · Authors · 2024-11-22
> **Addendum to Question 4**
>
> In question 4, we missed addressing the necessity of all loss functions during training if we use PIP's second stage PD controller and motion optimizer.
> When using PIP's second stage, the "garbage in, garbage out" principle applies: the quality of the first-stage estimate directly influences the second-stage estimate. Therefore, we want our first stage to be as good as possible. The physics-based losses play an essential role in this process by ensuring realistic motion. Without incorporating physics, motion estimates can become ambiguous, particularly in sparse sensor configurations.
> Furthermore, PIP uses ground-contact probablities, which they learn supervisedly. Our ground contact model is a spring-damper model, and in order to learn foot-ground interaction from IMU data alone, we need physical assumptions. Therefore, data tracking ($\mathcal{L} _ {\text{IMU}}$), physics ($\mathcal{L} _ {\text{K}}$, $\mathcal{L} _ {\text{T}}$) and ground contact assumptions ($\mathcal{L} _ {\text{slide}}$) are all necessary. Only $\mathcal{L} _ {tao}$ is optional with PIP's second stage, as its main effect is to reduce jitter and JTE. We show that the other losses have a positive effect on the model's performance in the ablation study.

---

> > ### Comment · Reviewer_SLE6 · 2024-11-24
> > **Response to the Authors, Part 1 (after submission of new experiments)**
> >
> > Thank you for the detailed response and for conducting additional experiments. I have read other reviews and your responses carefully, and I look forward to any new insights you can provide. I appreciate your comments for addressing my concerns on computational efficiency and evaluations. Now, I better understand where the paper stands compared to other works. I will provide my new reviews in the following comments. Please feel free to correct me if I am mistaken in my understanding of the paper. Thank you.
> >
> > **Decision:** First, I would like to adjust my review to *marginally above the acceptance threshold* but I want to note that it should come down to the expectations of the conference. **If the focus is on gait estimation research, then a weak acceptance is justifiable.** However, **if the emphasis is on groundbreaking novelty, a weak reject is more appropriate.** I have included my revised review below.
> >
> > * **Contributions:** While the paper effectively demonstrates the benefits of incorporating physics-based constraints and objectives in a self-supervised framework, this approach has been explored in other fields, such as motion reconstruction [1] and cloth simulation [2]. However, its applications within the gait estimation domain with IMUs contribute to the overall work in the gait analysis domain.
> >
> > * **Broader Impact on future works:** Some other works, such as [3] and [4], address a broader problem of solving 3D human motions using physics constraints, similar to the methodology used in this work. Although the problem formulation is different (video-based pose estimation, character control), more advanced physical constraints and 3D estimation methodologies have been explored in those works, which makes it difficult to judge the outcomes of this research for future works.
> >
> > * **Novelty:** I believe the main novelty of the paper is its self-supervised approach to developing a real-time solution for gait analysis without requiring labeled data. In self-supervised research, the network and objective design are often innovative and unique to the problem. However, they usually require large-scale datasets to be generalizable to different actions/noise levels and introduce additional challenges and limitations, which have not been explored extensively.
> >
> > * **Limitations:** The new and supplementary results show that the proposed method still has some challenges. I agree that it addresses the need for labeled data (regression/supervised models) and is faster than optimization-based solutions. However, more criteria should be considered when selecting the best model for a problem. For instance, inference time, training time, generalization to unseen data/actions, training noise robustness, inference noise robustness, validation error, and generalization with limited training data should be considered. I believe the regression/supervised models can be the most accurate (with enough training data), best for noise robustness (with enough noise considerations and data augmentations), and the fastest. In contrast, optimization-based approaches are slow and require heavy parameter tuning, but they can generalize very well to unseen data (in the absence of noise). This is why recent papers such as [5] and [6] propose to compare both approaches to obtain the most accurate and smooth results without sacrificing too much computational time. Based on the responses so far, my impression is that the proposed method lies somewhere in-between, where it does not require any labeled data, is as fast as regression-based models, and obtains better results than optimization-based models on in-distribution (training/inference on the same data/actions). However, as shown in the new experiments, it performs worse than regression-based models, might be sensitive and overfit to the input noise during training/testing (not tested), may or may not work on unseen data/actions (not tested), and it may require heavy parameter tuning (not tested, but the discussion on new experiments suggests that parameter-tuning is needed for different dataset sizes). Additionally, please correct me if I am wrong, for every new subject or action, the proposed method may need a re-calibration/re-training phase to adjust to the input signals before it can be deployed for real-time inference, which is not a consideration in most regression-based models. I should also mention that these are only my understanding of this space so far, and more detailed future research might be needed to verify these claims. Therefore, I do not expect the paper to change much based on this analysis, and it would not affect my ratings anymore. **TL;DR:** However, since understanding the limitations is a crucial point of any research, I strongly encourage the authors to discuss the advantages and shortcomings of each approach in the paper to be useful for future researchers in this field.

---

> > > ### Comment · Reviewer_SLE6 · 2024-11-24
> > > **Response to the Authors, Part 2 (after submission of new experiments)**
> > >
> > > * **Other Reviews:** It seems that other reviewers have also pointed out the limited evaluations, limited 2D-only experiments, and limitations for flat-ground assumptions. Other papers, specifically in physics-based 3D character control, already address these problems. Therefore, I have limited my expectations based on the gait estimation field of research. However, the opinions of others are still valid, and I would agree that the paper does not have enough impact.
> > >
> > >
> > > ### References
> > > - [1] Rempe, Davis, et al. "Humor: 3d human motion model for robust pose estimation." Proceedings of the IEEE/CVF international conference on computer vision. 2021.
> > >   - [1] uses physical constraints during training their CVAE (called regularization loss), but they compute the constraints differently from the proposed approach. Although this paper is not strictly on self-supervised research, it shows the applications of physical constraints during model training. Other research following this paper also uses similar objectives.
> > > - [2] Santesteban, Igor, Miguel A. Otaduy, and Dan Casas. "Snug: Self-supervised neural dynamic garments." Proceedings of the IEEE/CVF Conference on Computer Vision and Pattern Recognition. 2022.
> > >   - [2] is an example of physics-based self-supervised research on cloth deformation simulation.
> > > - [3] Xie, Kevin, et al. "Physics-based human motion estimation and synthesis from videos." Proceedings of the IEEE/CVF International Conference on Computer Vision. 2021.
> > > - [4] Tessler, Chen, et al. "Maskedmimic: Unified physics-based character control through masked motion inpainting." ACM Transactions on Graphics (TOG) 43.6 (2024): 1-21.
> > >   - Unlike the proposed model, [3] and [4] are not real-time and use physical constraints and dynamics during 3D character control.
> > > - [5] Yi, Xinyu, et al. "Physical inertial poser (pip): Physics-aware real-time human motion tracking from sparse inertial sensors." Proceedings of the IEEE/CVF conference on computer vision and pattern recognition. 2022.
> > > - [6] Kolotouros, Nikos, et al. "Probabilistic modeling for human mesh recovery." Proceedings of the IEEE/CVF international conference on computer vision. 2021.
> > >   - [5] and [6] are examples of two-stage (regression + optimization) network designs

---

> > > > ### Author Response · Authors · 2024-11-25
> > > > **Response to Reviewer SLE6**
> > > >
> > > > Thank you very much for your insights. Your comment has a lot of interesting points, and we agree with most of them. For a few specific aspects, we’d like to share some additional thoughts:
> > > >
> > > > **Contributions / Broader Impact**:
> > > > We agree that the idea to turn physical simulations into learning objectives is not new in itself, but new to inertial gait analysis and that this is a good positioning for the paper. For future work, we are also looking into 3D inertial pose estimation and learning movement controllers with the same core idea.
> > > >
> > > > **Limitations**:
> > > > > I believe the regression/supervised models can be the most accurate (with enough training data), best for noise robustness (with enough noise considerations and data augmentations), and the fastest.
> > > >
> > > > We largely agree. However, regarding "best for noise robustness," we believe that self-supervised systems can achieve similar robustness, provided the same noise considerations are applied. While this hasn’t been fully implemented in our current work, we are introducing Gaussian noise to the input signals as a form of data augmentation, and we believe additional measures could bridge any gap in robustness.
> > > >
> > > > > Additionally, please correct me if I am wrong, for every new subject or action, the proposed method may need a re-calibration/re-training phase to adjust to the input signals before it can be deployed for real-time inference, which is not a consideration in most regression-based models.
> > > >
> > > > - New actions: Regression-based models would also require calibration when encountering out-of-distribution actions. For example, this is illustrated in the video at 6:03 on PIP’s GitHub page [1].
> > > >
> > > > - New subjects: Recalibration is not necessarily required for new participants in our approach. As mentioned in the “General Discussion” section of the paper, the method performs similarly on training and test participants in the experiment “Ours - [1].” When the participants' height is out-of-distribution, however, the ground contact model will not produce realistic GRF values. We could also frame that as an advantage: The model can be personalized.
> > > >
> > > > > and limitations for flat-ground assumptions.
> > > >
> > > > In principle, the information about ground incline / stairs is present in the IMU data, therefore it can potentially be modeled. In our point of view, this is a bit far-fetched and a future challenge.
> > > >
> > > > [1] https://xinyu-yi.github.io/PIP/

---

### Official Review · Reviewer_ig7e · 2024-11-06

**Soundness:** 3
**Presentation:** 2
**Contribution:** 3
**Rating:** 5
**Confidence:** 1

**Summary:**

This paper presents a method to estimate the human pose and dynamics from wearable IMU sensors. Specifically, a self-supervised method is used to reconstruct the input IMU signals and to estimate joint angles and dynamics. Kane’s equations are used to calculate the motion equations. Experiments show promising performances on dynamics and pose estimations.

**Strengths:**

1. Estimating human dynamics from IMU sensors is an important and meaningful research direction.

2. Using a self-supervised learning method to capture information from IMU sensors sounds very interesting.

3. Combining a neural network with a physics model is also reasonable.

**Weaknesses:**

1. In line 199 it is mentioned that "the IMU data consists of 2D acceleration and 1D gyroscope data per sensor". Why is the acceleration 2D and gyroscope data 1D, instead of 3D? Besides, what does the "gyroscope measurement" refer to? Is it angular velocity or estimated orientation?

2. I am also a little bit confused by the proposed method. It is mentioned that this is a self-supervised method, however, in Section 3.1, the output contains root rotation and translations, joint angles and joint torques. I don't quite understand how each of these outputs is predicted (e.g., what is estimated by neural networks, what is calculated from Kane’s equations)

3. Besides, are the motion capture data used for this method? or purely based on IMU data (since it is self-supervised)? If ground truth joint angles are provided to supervise training, it will not be self-supervised learning.

4. I can not find much information about the adopted dataset Dorschky et al., 2024. I am curious about the choice of dataset and suggest using some well established dataset like [1,2].

[1] Zhang et al.: MMVP: A Multimodal MoCap Dataset with Vision and Pressure Sensors, CVPR 2024

[2] Werling et al.: AddBiomechanics Dataset: Capturing the Physics of Human Motion at Scale, ECCV 2024

5. I think it would be very important and beneficial to show the proposed method can benefit the existing 3D pose estimation methods from IMU sensors. However, such a comparison is only provided in the appendix using different datasets, and compared to SMPL models, the joint angle ranges are manually set. There are also recent works that try to make SMPL models or AMASS datasets align with biomechanical models with a meaningful degree of freedom, like [3]. Perhaps similar methods can be applied to enable a fair comparison.

[3] MANIKIN: Biomechanically Accurate Neural Inverse Kinematics for Human Motion Estimation, ECCV 2024

**Questions:**

See above

---

> ### Author Response · Authors · 2024-11-18
> **Response to Reviewer ig7e**
>
> Dear Reviewer ig7e,
>
> thank you for your feedback. Currently, we are preparing additional experiments and will provide a revised version of the paper when finished. We will address your comments in the following way in the revised version:
>
> ## Questions:
>
> > In line 199 it is mentioned that "the IMU data consists of 2D acceleration and 1D gyroscope data per sensor". Why is the acceleration 2D and gyroscope data 1D, instead of 3D? Besides, what does the "gyroscope measurement" refer to? Is it angular velocity or estimated orientation?
>
> The IMU data consists of 3D acceleration and 3D gyroscope data per sensor. We only consider the saggital plane, where the acceleration is 2D (x-axis and y-axis) and the gyroscope data is 1D (z-axis). The gyroscope measurement refers to the angular velocity.
>
> > I am also a little bit confused by the proposed method. It is mentioned that this is a self-supervised method, however, in Section 3.1, the output contains root rotation and translations, joint angles and joint torques. I don't quite understand how each of these outputs is predicted (e.g., what is estimated by neural networks, what is calculated from Kane’s equations)
>
> The neural network predicts the root rotation and translations, joint angles, and joint torques as well as their first and second derivatives directly. Next, the ground reaction forces is estimated from the kinematics of the ankle joint, which is also predicted by the neural network. Kane's method is only used during training as a loss function to ensure the physical consistency of the predicted outputs and the estimated ground reaction forces. In summary, all evaluated outputs except the ground reaction forces are directly predicted by the neural network.
>
> > Besides, are the motion capture data used for this method? ...
>
> The motion capture data is only used during evaluation.
>
> > I can not find much information about the adopted dataset Dorschky et al., 2024. I am curious about the choice of dataset and suggest using some well established dataset like [1,2].
>
> Thank you for the suggestion of these datasets. Unfortunately, both datasets do not contain IMU data. Reviewer Nikt has pointed us to dataset [1], which we are currently investigating. We will include the dataset in the revised version of the paper. The current dataset, Dorschky et al., 2024, contains IMU of the main modes of locomotion in the sagittal plane, walking and running, which is why it is suitable for our study. Furthermore, it contains measured positions of the IMUs on the body, which we deemed necessary for the proposed method.
>
> [1] Camargo, J. et al. "A comprehensive, open-source dataset of lower limb biomechanics in multiple conditions of stairs, ramps, and level-ground ambulation and transitions." J Biomech, 2021.
>
> > I think it would be very important and beneficial to show the proposed method can benefit the existing 3D pose estimation methods from IMU sensors. ...
>
> We agree with you that a self-supervised 3D pose estimation method would be very beneficial. However, as the conceptually very similar optimal control problems have not been successfully solved for the 3D case, we decided to focus on the 2D case. Further challenges could be a significantly longer training time and possible higher chance of local minima. We leave that as an open challenge for future work.
>
> > ...However, such a comparison is only provided in the appendix using different datasets, and compared to SMPL models, the joint angle ranges are manually set. There are also recent works that try to make SMPL models or AMASS datasets align with biomechanical models with a meaningful degree of freedom, like [3]. Perhaps similar methods can be applied to enable a fair comparison.
>
> A fair comparison to 3D pose estimation models would be interesting. In order to be kinematically consistent with biomechanical data, we agree that more biomechanically accurate models like MANIKIN, SKEL [2] or OpenSim [3] are needed. We use costum joint ranges. Joint ranges differ from person to person, therefore, we wanted to keep those as unbounded as possible in our experiments. Therefore, we imposed joint ranges of 120° of flexion and extension in our training, except for knee extension, which was set to ca 6°. Bounds on joint ranges are needed in the first place as the rotation can be ambiguous, for example, all other losses would treat a joint angle of $2\pi$ equally to a joint angle of 0. For the knee, we set the range to a slight overextension, so that this joint angle is also not reached in practice. When using the same restrictions for the knee as for the other joints, the model would occasionally land in local minima, where large knee overextensions are predicted.
>
> [2] Keller, M. et al. "Towards Biomechanically Accurate 3D Digital Humans", SIGGRAPH ASIA 2023.
>
> [3] Delp, S. et al. "OpenSim: Open-source software to create and analyze dynamic simulations of movement", IEEE TBME, 2007.

---

### Author Response · Authors · 2024-11-22
**General Discussion - New Experiments**

Dear Reviewers,

Thank you for your valuable feedback. We have conducted additional experiments to address your comments. In this thread, we will present the results of these experiments and discuss their implications. Some experiments are still running - we keep this thread updated.

### Comparable Testing

As mentioned by Reviewer SLE6, we have compared our method with baseline methods [1] and [2] on the same evaluation dataset. To enable a fair comparison, we have altered the training and evaluation schemes of our method to match those of [1] and [2].

In [1], the authors trained a CNN on ensemble averaged gait cycles from participants 4-10 as well as simulated gait cycles. They evaluated on ensemble averaged gait cycles from participants 1-3. To compare our method with [1], we trained "Ours - [1]" on the raw data of participants 4-10 and evaluated on ensemble averaged gait cycles from participants 1-3. Therefore, our training scheme is different from [1], but the evaluation scheme is the same. This way, we can train our model without further preprocessing steps.

In [2], the authors solved optimal control problems on averaged gait cycles for each trial individually. To avoid the need for retraining, we trained and evaluated "Ours - [2]" on ensemble averaged gait cycles for all participants and trials. As our approach is fully self-supervised, there is no label leakage in doing so.

_Table 1: Comparison of our method with [1,2] on the same evaluation dataset._

| Model | JAE [°] | JTE [BWBH%] | GRF [BW%] | Speed [m/s] |
| --- | --- | --- | --- | --- |
| Ours - [1] | 11.2 | 7.2 | 23.1 | 0.12 |
| [1] | 4.9 | 1.4 | 10.7 | - |
| --- | --- | --- | --- | --- |
| Ours - [2] | 8.9 | 6.8 | 23.8 | 0.25 |
| [2] | 6.3 | 2.6 | 17.9 | 0.25 |


For comparison, when testing "Ours - [1]" on emsemble averaged gait cycles from participants 4-10 (training participants), we obtained the following results:
JAE: 11.24°; JTE: 7.05 BWBH%; GRF: 24.3 BW%; Speed: 0.15 m/s. "Ours - [1]" was trained on the raw IMU streams from participants 4-10, and exhibits a drop-off in JAE and JTE when tested on ensemble averaged gait cycles, but not when tested on different participants. Therfore, we conclude that the model is generalizing well over participants. On the other hand, the domain shift from raw IMU streams to ensemble averaged gait cycles does impact the performance. Lastly, the speed error is decreased in this evaluation setting.

For "Ours - [2]", there is no domain shift, but the training data itself only consists of 60 gait cycles. Therefore, we expect that this training run did not generalize as well as the other models. We set $\lambda _ {\text{FS}}$ to 30, as we found the amount of training data to be problematic.

[1] Dorschky et al. "CNN-Based Estimation of Sagittal Plane Walking and Running Biomechanics From Measured and Simulated Inertial Sensor Data", Frontiers in Bioengineering and Biotechnology, 2020.

[2] Dorschky et al. "Estimation of gait kinematics and kinetics from inertial sensor data using optimal control of musculoskeletal models." Journal of Biomechanics, 2019.

---

> ### Author Response · Authors · 2024-11-25
> **General Discussion - New Experiments #2**
>
> ### Testing on other motions
>
> We have tested SSPINNpose on a dataset containing a variety of motions, for example jumps, lunges, and squats [1]. In the supplementary material, we show the video results of these motions. We found that SSPINNpose can learn these motions from IMU data alone and produces realistic movements. We trained on 7 participants from the first data aquisition phase and show motions on the remaining 3 participants. However, we found it challenging to use the dataset with our approach, as some challenges occured that make the results less accurate than for walking and running. These challenges would require additional care or preprocessing steps, and for these reasons, we will not show results on this dataset in the main paper:
>
> - **Missing pelvis IMU**: The dataset contains IMU data for the feet, shanks, and thighs, but not for the pelvis. For motions that involve significant torso tilt, the torso orientation remains ambiguous, and therefore the hip angle erroneous.
> - **Missing participant information**: The dataset does not contain information about the participants' height or IMU placements. For the thigh and shank IMUs, we were able to guess these from the figures in the publication. The foot-worn IMU, however, is not described and we guessed a placement in the mid-foot, which is likely not accurate. It could be that the foot-worn IMU (which is part of the insole) is placed at the heel or the toe, which could affect the results when the shoe bends during the motion. In our results, you will see that the ankle angle is strongly biased towards plantarflexion.
>
> The dataset that we originally intended to evaluate, [2], only contains IMU data for a single leg, making it unsuitable for our method. We have searched for further datasets that contain _a)_ IMU data for both legs, _b)_ ground truth joint angles and joint moments, _c)_ measured IMU placements, and _d)_ a variety of movements in the sagittal plane, but did not find a suitable dataset.
>
>
> [1] Molinaro et al. "Task-agnostic exoskeleton control via biological joint moment estimation". Nature, 2024.
>
> [2] Estimation of Lower Limb Joint Angles and Joint Moments during Different Locomotive Activities Using the Inertial Measurement Units and a Hybrid Deep Learning Model. Sensors 23.22 (2023): 9039.
>
> Edit: the motion clips are in the folder "molinaro2024/" | Edit2: Removed background audio from videos

---

> ### Author Response · Authors · 2024-11-26
> **General Discussion - New Experiments #3**
>
> ### Sensitivity to IMU misplacements
>
> Reviewer CE42 asked about the sensitivity of our method to variations in IMU placement. We have conducted following experiment to investigate this:
>
> _Uncertain IMU positions in training:_ We retrained SSPINNpose with IMU positions that were perturbed by varying offsets in the sagittal plane. We did a training run each with offsets of 2 cm, 4cm, 6 cm, 8 cm, 10 cm, 15 cm, and 20 cm. The results compared to our baseline experiment (no offsets) are shown in Table 2.
>
> _Table 2: Sensitivity of SSPINNpose to IMU placement._
> | Offset [cm] | JAE [°] | JTE [BWBH%] | GRF [BW%] | Speed [m/s] |
> | --- | --- | --- | --- | --- |
> | Baseline (0) | 8.9 | 5.0 | 18.8 | 0.15 |
> | 2 | 9.4 | 5.5 | 19.1 | 0.13 |
> | 4 | 9.7 | 5.9 | 19.5 | 0.14 |
> | 6 | 8.4 | 5.1 | 17.6 | 0.13 |
> | 8 | 10.4 | 5.7 | 20.9 | 0.16 |
> | 10 | 16.4 | 4.7 | 36.1 | 2.99 |
> | 15 | 14.1 | 6.0 | 22.1 | 0.52 |
> | 20 | 18.5 | 5.3 | 19.7 | 0.39 |
>
> The results indicate that SSPINNpose is robust to small variations in IMU placement, but performance degrades with offsets of 10 cm or more. Manual IMU placement or measurement should usually be accurate within less than 10 cm, so we consider this a reasonable level of robustness.
>
> Edit: Error when copying the GRFE in the initial version of this comment

---

> > ### Author Response · Authors · 2024-11-27
> > **General Discussion - New Experiments #4**
> >
> > _Uncertain IMU positions for finetuning:_ We fine-tuned the models from the "sensitivity to IMU placement" experiment results to estimate the true IMU placement. We compare these results to the original finetuning experiment and the IMU position from the dataset in Table 3. As in the original finetuning experiment, we did not estimate the IMU placement for the pelvis IMU, as there are not enough pelvic tilt motions in the dataset to properly estimate the placement. The results show that the model is able to partially recover erroneous IMU placements, as long as the model has been trained on reasonably accurate IMU placements.
> >
> > | Offset [cm] | Offset to Sec. 4.2. +- std [cm] | Offset to IMU placement in dataset +- std [cm] |
> > | --- | --- | --- |
> > | Baseline (0) | 0.0 +- 0.0 | 4.7 +- 2.2 |
> > | 2 | 4.4 +- 2.0 | 2.2 +- 1.2 |
> > | 4 | 6.1 +- 2.5 | 4.3 +- 2.5 |
> > | 6 | 3.8 +- 2.3 | 2.1 +- 1.4 |
> > | 8 | 5.8 +- 2.3 | 4.1 +- 1.8 |
> > | 10 | 7.6 +- 3.3 | 6.3 +- 3.7 |
> > | 15 | 15.1 +- 8.7 | 14.5 +- 9.9 |
> > | 20 | 24.1 +- 11.6 | 23.7 +- 12.5 |

---

### Author Response · Authors · 2024-11-28
**Revised version of manuscript uploaded**

Dear Reviewers,

thank you for your insights and helpful suggestions. Your comments contributed to a significant improvement, in our opinion. We have uploaded a revised PDF earlier. In summary, we have made the following changes and additions:

1. **Clearer positioning of our work**: We acknowledge that our work is related to self-supervised physics-based methods from other domains, suchs as computer vision and graphics. These similarities are laid out in a new section in the related work section.

2. **Contribution**: Our contribution is now more clearly stated in the introduction.

3. **Deep-learning-based methods**: As suggested, we have expanded our related work section on deep-learning-based methods. This section now includes a list of methods that use two-step inference in computer vision and graphics, as well as single-step regression methods that stem from biomechanical applications.

4. **New Experiments**: As already laid out in the discussion thread, we have conducted a number of new experiments to validate our claims. These experiments include:
    - A fairer comparison with baseline methods on the same evaluation scheme.
    - Sensitivity to IMU misplacements.
    - Recovery of erroneous IMU placements.

    Furthermore, we conducted two experiments that are only briefly or not mentioned in the paper:
    - Testing on a different dataset containing a variety of motions. Here, we encountered challenges with the dataset that make our method only applicable when taking additional assumptions that might not be accurate.
    - Combination with PIP's second stage. Here, we were unable to show a considerable improvement over our baseline method, however, that might be because our model has already generalized well to the tested motions.

5. **Minor Additions**: In Section 4.1, we disussed some of the upsides and downsides of our method compared to two-stage dynamics inference methods. Based on your comments, we have added small remarks and clarifications for better understanding throughout the paper.

We believe these changes address your concerns and significantly enhance the manuscript's quality. We look forward to engaging further in the discussion phase. Thanks again for your feedback.

---

### Meta-Review · Area_Chair_cYXs · 2024-12-24

**Metareview:**

## Summary
The paper introduces SSPINNpose, a self-supervised method for estimating human pose and dynamics from wearable IMU sensors. The method reconstructs input signals and estimates joint angles and dynamics using Kane's equations. Experiments show promising results in dynamics and pose estimations. SSPINNpose is a physics-informed neural network that accurately estimates joint angles and moments in real-time, even with sparse sensor configurations. It can infer sensor positions and achieves comparable performance to supervised methods in estimating lower-body dynamics during walking and running at various speeds.

## Strengths
* Highlights the importance of human dynamics estimation from IMU sensors and the use of auxiliary physics-inspired losses for realistic estimations.
* Suggests using self-supervised learning to capture information from IMU sensors by combining a neural network with a physics model for gait kinematic estimation.
* Demonstrates real-time capability and physical plausibility of the method.
* Shows versatility in handling sparse IMU setups and inferring sensor placements.

## Weaknesses
* The authors' approach lacks comparisons with state-of-the-art methods and for not providing direct comparisons with other evaluation schemes.
* The paper's contributions are not elaborated on, and the computational cost of adding all objective functions during training is evaluated.
* The authors' approach is limited to 2D movements and does not address full 3D motion estimation.
* The method assumes flat ground and non-sliding feet, which may affect the accuracy of estimations in real-world scenarios.
* The authors use a generic multibody dynamics model without personalization, potentially affecting accuracy due to individual anatomical differences.
* The main self-supervised objective is minimizing the deviation between actual IMU motion and the virtual model’s motion, which closely resembles a supervised objective.
* The authors miss many common regression-based works that estimate joint kinematics regression of one joint from other inputs.

## Conclusions
Based on the reviews and author’s feedback, the paper has potential to be accepted, however there were some concerns about the paper as stated by two reviewers that were not convinced at all. For instance, as mentioned by one of the reviewers *It seems that other reviewers have also pointed out the limited evaluations, limited 2D-only experiments, and limitations for flat-ground assumptions. Other papers, specifically in physics-based 3D character control, already address these problems. Therefore, I have limited my expectations based on the gait estimation field of research. However, the opinions of others are still valid, and I would agree that the paper does not have enough impact.* and other reviewer had some concerns about the definition of self-supervised learning and the amount of data. Therefore, the paper should address all the concerns before accepting the paper.

**Additional Comments On Reviewer Discussion:**

The summary is described in the conclusions of the previous section.

---

### Decision · Program_Chairs · 2025-01-22

Reject